# Long-Short Alignment for Effective Long-Context Modeling in LLMs

Tianqi Du [* 1]  Haotian Huang [* 2]  Yifei Wang [3]  Yisen Wang [1 4]

## Abstract

Large language models (LLMs) have exhibited impressive performance and surprising emergent properties. However, their effectiveness remains limited by the fixed context window of the transformer architecture, posing challenges for long-context modeling. Among these challenges, length generalization—the ability to generalize to sequences longer than those seen during training—is a classical and fundamental problem. In this work, we propose a fresh perspective on length generalization, shifting the focus from the conventional emphasis on input features such as positional encodings or data structures to the output distribution of the model. Specifically, through case studies on synthetic tasks, we highlight the critical role of **long-short alignment**—the consistency of output distributions across sequences of varying lengths. Extending this insight to natural language tasks, we propose a metric called Long-Short Misalignment to quantify this phenomenon, uncovering a strong correlation between the metric and length generalization performance. Building on these findings, we develop a regularization term that promotes long-short alignment during training. Extensive experiments validate the effectiveness of our approach, offering new insights for achieving more effective long-context modeling in LLMs. Code is available at https://github.com/PKU-ML/LongShortAlignment.

## 1. Introduction

Large language models (LLMs) have demonstrated impressive abilities in various tasks such as natural language generation, reading comprehension, code synthesis, instruction-following, and commonsense reasoning (Radford et al., 2019; Brown et al., 2020; Chowdhery et al., 2023; Touvron et al., 2023). Their performance has consistently improved by scaling both model and dataset sizes (Kaplan et al., 2020). However, the effectiveness of LLMs remains limited by the fixed context window of the Transformer architecture, posing significant challenges for *long-context modeling*. With larger context sizes, a model can benefit from more in-context learning examples, a greater number of reasoning steps, or the ability to generate longer coherent texts (Li et al., 2024; Huang & Chang, 2023; Wang et al., 2024b). Nevertheless, training a Transformer with long input sequences is often prohibitively slow and memory-intensive, making it crucial to understand and improve how LLMs generalize to longer contexts.

A classical and foundational subproblem within long-context modeling is *length generalization*—the ability to generalize from shorter training sequences to longer test sequences (Anil et al., 2022). This remains a major challenge even for large-scale Transformers (Liu et al., 2024). Understanding the mechanisms of length generalization is thus an essential step toward achieving robust and efficient long-context modeling.

There exist two dominant approaches to understanding and improving length generalization. The first is to design better positional encodings (PE) (Press et al., 2021; Su et al., 2024; Kazemnejad et al., 2023; Peng et al., 2024; Chen et al., 2024; Yang, 2023; Zhang et al., 2024b), which help the model systematically encode tokens across a wide range of positions. By reducing the inductive gap between short training sequences and long test sequences, these encodings can partially improve generalization (Kazemnejad et al., 2023). The second is to analyze the underlying mechanisms of Transformer models (Zhou et al., 2024; Lee et al., 2023; Veličković & Blundell, 2021; Nogueira et al., 2021; Deletang et al., 2022), such as what algorithms they can simulate or how task design affects generalization. However, we observe that one key aspect has been largely overlooked: the *output space* of the model. In this work, we argue that output behavior plays a central role in determining generalization quality across context lengths.

We begin our analysis with synthetic tasks involving length

---
[*]Equal contribution [1]State Key Lab of General Artificial Intelligence, School of Intelligence Science and Technology, Peking University, China [2]NUS, Singapore [3]MIT CSAIL, USA [4]Institute for Artificial Intelligence, Peking University, China. Correspondence to: Yisen Wang <yisen.wang@pku.edu.cn>.

*Proceedings of the 42nd International Conference on Machine Learning*, Vancouver, Canada. PMLR 267, 2025. Copyright 2025 by the author(s).

generalization: predicting the mean value and the length of binary sequences. Both empirical and theoretical results reveal a stark contrast—Transformers generalize well on the mean prediction task but struggle on the length prediction task. The key difference lies in the support set of the output distribution: in the mean prediction task, it remains stable across input lengths, while in the length prediction task, it varies with the sequence length. We hypothesize that this **misalignment in the output distribution leads to poor generalization** in the latter task. To verify this, we propose a reparameterization technique named OutRep that explicitly aligns the output distributions across lengths. Our analyses confirm that this approach significantly improves generalization, supporting our hypothesis.

Building on this insight, we extend our findings to natural language tasks. Although natural language outputs are vector-valued and more complex than scalar outputs in synthetic tasks, similar misalignment phenomena arise. For instance, two sequences with the same ending but slightly different lengths should ideally produce similar output distributions. However, models that generalize poorly often produce divergent outputs for such inputs. To quantify this phenomenon, we introduce a metric called **long-short misalignment**, which measures the divergence in output distributions via symmetric cross-entropy. Both empirical and theoretical analyses show that this metric correlates strongly with long-context performance—more so than traditional training loss—making it a reliable indicator of generalization. Motivated by this, we incorporate the long-short misalignment metric as a regularization term during training. Extensive experiments across both synthetic and natural language tasks validate the effectiveness of this approach.

The main contributions of this work are as follows:

- We identify the crucial role of output behavior in long-context modeling, with a focus on length generalization. Both empirical and theoretical evidence shows that misalignment in output distributions across input lengths leads to poor generalization.

- We introduce the **long-short misalignment** metric to quantify this output discrepancy and demonstrate its strong correlation with length generalization ability.

- We incorporate this metric into a novel regularization term. Extensive experiments validate the effectiveness of this approach in boosting long-context modeling performance.

## 2. Related Work

**Length Generalization on Synthetic Tasks.** Our paper is related to the line of work that seeks to understand the capabilities and limitations of Transformer models when

it comes to algorithmic reasoning (Veličković & Blundell, 2021). Specifically, we focus on simple tasks and study length generalization on the standard Transformer architecture with causal structure. Related to this, Lee et al. (2023) study how well transformers trained from scratch can learn simple arithmetic tasks, and find that no length generalization is observed. Nogueira et al. (2021) find that partial length generalization on addition is observed only when models reach 3B parameters and when the addition questions are presented in reverse order. Zhou et al. (2024) proposes the RASP Generalization Conjecture that Transformers tend to learn a length-generalizing solution if there exists a short RASP-L program that works for all input lengths. Liu et al. (2023) discovers that the Transformer will learn shortcuts through the study of various synthetic tasks. Besides these explorations on specific tasks, some works study the impact of different positional encodings on the length generalization of math reasoning tasks (Press et al., 2021; Ontanon et al., 2022; Kazemnejad et al., 2023; Ruoss et al., 2023). More details of these works can be viewed in Appendix A.

**Long-context Modeling on Natural Language Tasks.** A series of works (Sun et al., 2023; Chi et al., 2022; 2023; Zhang et al., 2024b; Chen et al., 2024; Peng et al., 2024; Yang, 2023; Chen et al., 2023a; Fang et al., 2025b) aim to extend the context size of Transformer-based models during fine-tuning, primarily by modifying positional encodings. For example, Zhang et al. (2024b) introduces a novel extension to RoPE (Su et al., 2024) which combines adjusting RoPE's base frequency and scaling the attention logits to help LLMs efficiently adapt to a larger context window. Chen et al. (2024) generalizes the positional encoding scaling approaches to model the continuous dynamics by ordinary differential equations over the length scaling factor. Wang et al. (2024a) proposes a novel approach designed to narrow the generalization gap and provides length extrapolation analysis on the feature gap. Our approach, in contrast, focuses on the model's output space, which identifies the crucial role of long-short alignment in length generalization.

## 3. A Case Study on Synthetic Tasks: How Long-Short Alignment Affects Length Generalization?

Long-context modeling aims to extend the reasoning and generation capabilities of language models to longer input sequences (Fang et al., 2025a; Wu et al., 2025; Kuratov et al., 2024; Zhang et al., 2024b). One fundamental challenge in this setting is length generalization—the ability to generalize from shorter training contexts to longer ones. Prior work has identified several factors that influence length generalization, including the task type (Zhou et al., 2024; Jelassi

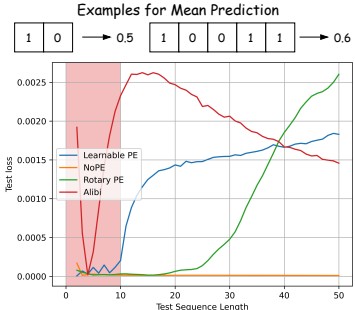

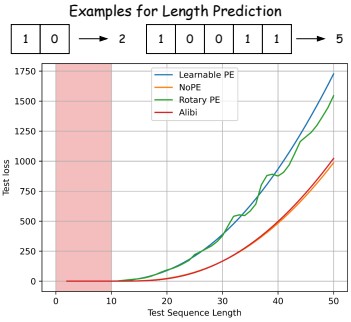

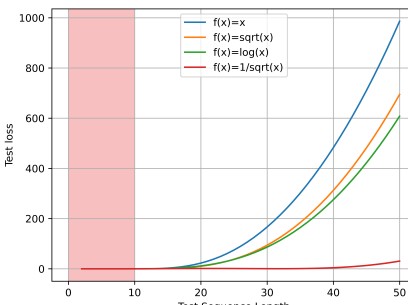

(a) Length generalization in the **mean** prediction task with different positional encodings

(b) Length generalization in the **length** prediction task with different positional encodings

(c) Length generalization in the **length** prediction task with different reparameterization function.

Figure 1: Comparison between the length generalization performance in the mean prediction task and the length prediction task. The training sequence length is uniformly selected from $[1, 10]$ (indicated by the light red area) while the test sequence lengths (on the x-axis) can reach a maximum of 50. In the length prediction task (b), the model struggles with length generalization. Conversely, the model demonstrates significantly better length generalization in the mean prediction task (a) using NoPE (indicated by the orange line). Figure (c) shows the length generalization performance in the length prediction task using different reparameterization functions $f(x)$. All three reparameterized targets improve generalization compared to the origin (blue) target. Among them, $f(x) = 1/\sqrt{x}$ (red) performs exceptionally well.

et al., 2023; Nogueira et al., 2021) and the design of positional encodings (Ontanon et al., 2022; Kazemnejad et al., 2023; Ruoss et al., 2023). In this work, we propose a fresh perspective by examining the **consistency of the model's output distribution** across inputs of varying lengths, named long-short alignment, identifying it as a crucial yet underexplored factor for effective long-context modeling.

**Mean Prediction v.s. Length Prediction.** We start from a case study on synthetic tasks: in the mean prediction task, the prediction target is the mean value of the sequence, while in the length prediction task, the target is the length of the sequence. We focus on binary input sequences, where each position in the sequence is filled with 0 or 1 with the same probability, and the decoder-only Transformer (Vaswani et al., 2017), a model widely used in both synthetic tasks (Zhou et al., 2024; Jelassi et al., 2023) and LLMs (Touvron et al., 2023; Peng et al., 2024), which utilizes a causal mask in the self-attention module to enable auto-regressive generation. More model details can be found in Appendix B. We train the model on sequences with a maximum length of $l_{\text{train}} = 10$ and test it on sequences with a maximum length of $l_{\text{test}} = 50$. Figure 1a and Figure 1b display the test results for both tasks. We observe that regardless of the positional embedding used, the test loss of the length prediction task dramatically increases when the test sequence length exceeds 10, the maximum training length. Furthermore, the test loss continues to rise as the test sequence length grows, indicating that the model demonstrates very **low length generalization ability in the length prediction task.** In contrast, the model exhibits **strong generalization**

**ability in the mean prediction task**, as the test loss on longer sequences remains nearly consistent with the loss on shorter sequences. We now provide a theoretical analysis of this observation.

**Theorem 3.1.** *In the length prediction task, the length generalization loss $\mathcal{E}_{\text{length}}(\cdot; \cdot)$ has a quadratic relationship with the predicted length $l_{\text{test}}$, i.e.,*

$$\mathcal{E}_{\text{length}}(g_\theta^{l_{\text{train}}}; l_{\text{test}})$$
$$= \mathbb{E}_{\mathbf{x}_{\text{test}} \in \{0,1\}^{l_{\text{test}}}} \left[ \left\| g_\theta^{l_{\text{train}}}(\mathbf{x}_{\text{test}}) - y(\mathbf{x}_{\text{test}}) \right\|_2^2 \right] \quad (1)$$
$$= \mathcal{O}\left( (l_{\text{test}} - l_{\text{train}})^2 \right),$$

*where $g_\theta^{l_{\text{train}}}$ is the model trained on sequences with maximum training length $l_{\text{train}}$, $\mathbf{x}_{\text{test}}$ is the testing input with length $l_{\text{test}}$.*

*However, in the mean prediction task, the length generalization loss has a fixed upper bound:*

$$\mathcal{E}_{\text{mean}}(g_\theta^{l_{\text{train}}}; l_{\text{test}}) = \mathcal{O}(1). \quad (2)$$

The full statement and the proof are shown in Appendix C.1. From both the empirical and theoretical results, it is evident that while the mean and length of a sequence all convey global information, the model's length generalization ability varies across these tasks. A key distinction lies in the differences in **output distribution** for each task. In the mean prediction task, where the model generalizes well, the output remains within the fixed range of $[0, 1]$, regardless

of sequence length. However, in the length prediction task, where generalization is poor, the support set of the output distribution shifts to a single-point set $\{l\}$ as the sequence length increases to $l$. This distinction between the two types of tasks motivates us to consider the importance of long-short alignment for better length generalization ability.

**Explicit long-short alignment helps length generalization.** We propose output reparameterization (OutRep), a reparameterization technique to explicitly improve long-short alignment in synthetic tasks, thereby enhancing the model's length generalization ability. In the length prediction task, the output distribution for sequences of certain lengths is known. Leveraging this prior knowledge, during training, we apply a reversible function $f : \mathbb{R} \to \mathbb{R}$ to map the support sets of output distributions for sequences of varying lengths into more aligned sets. Instead of using the original target $y(\mathbf{x})$, we train the model on the transformed target $f(y(\mathbf{x}))$. At test time, we apply the reverse function $f^{-1}$ to the output to recover the original prediction. This approach aligns the output distributions across different lengths, which is expected to improve length generalization. We consider the following reparameterization functions: $f(x) = \sqrt{x}$, $f(x) = \log(x)$ and $f(x) = 1/\sqrt{x}$. We show the experiment results in Figure 1c. It can be observed that all three reparameterization functions successfully relieve the poor length generalization ability in the length prediction task. Specifically, the reparameterization function $f(x) = 1/\sqrt{x}$ has a nearly perfect generalization ability when the length is no more than 35. The rising trend when the test sequence length becomes longer is still slow. These results verify our conjecture on the long-short alignment that **better long-short alignment leads to improved length generalization ability**. We add more theoretical results and discussions in Appendix C and Appendix D. In the next section, we will extend these findings to the more practical natural language tasks.

# 4. Long-Short Alignment in Natural Language Tasks

In the previous section, we observed a positive correlation between length generalization ability and long-short alignment in synthetic tasks. Motivated by this finding, in this section, we aim to extend this investigation to natural language tasks. First, we introduce a metric to quantify long-short alignment in sequence modeling and demonstrate its strong correlation with performance on long-context benchmarks. Building on this insight, we propose incorporating this metric as a regularization term during training to improve long-short alignment, which can lead to the performance gains detailed in Section 5.

## 4.1. Long-Short Misalignment: Quantifying the Discrepancy of Output Distributions

In synthetic tasks, we measure the discrepancy between the support sets of output distributions to capture the differences in output across varying sequence lengths. However, in natural language tasks, the model output is a vector $g_\theta(\mathbf{x}) \in \mathbb{R}^{|\mathcal{V}|}$, where the dimension is the size of the vocabulary $|\mathcal{V}|$. This makes it challenging to directly apply the same analysis from synthetic tasks to natural language tasks. Despite this, similar long-short misalignment issues can still be observed in natural language tasks. Specifically, for a sequence $\mathbf{x}$ and its suffixes $\mathbf{x}_{[-l_1:]}$ and $\mathbf{x}_{[-l_2:]}$, where $l_1$ and $l_2$ are two lengths and $\mathbf{x}_{[-l_i:]}$ means the last $l_i$ tokens of $\mathbf{x}$ $(i = 1, 2)$, the model's output is expected to remain consistent when $l_1$ and $l_2$ are similar, because the two suffixes share large overlap in tokens, resulting in similar contextual information. However, we find that models with poor length generalization tend to produce distant output distributions when conditioned on these sequences.

### 4.1.1. METRIC

To quantitatively explore the relationship between long-short alignment and length generalization ability, we want to first design a metric to evaluate long-short alignment. We propose to utilize symmetrical cross-entropy (SCE) loss (Wang et al., 2019) to measure the divergence between output distributions conditioned on two distinct sequences. Consider two input sequences, $\mathbf{x}$ and $\mathbf{x}'$ with corresponding model predictions $\mathbf{y} = g_\theta(\mathbf{x})$ and $\mathbf{y}' = g_\theta(\mathbf{x}')$. The SCE loss between these predictions is defined as:

$$\mathcal{L}_{\mathrm{SCE}}(\mathbf{y}, \mathbf{y}') = - \left( \langle \mathbf{y}', \log(\mathbf{y}) \rangle + \langle \mathbf{y}, \log(\mathbf{y}') \rangle \right), \quad (3)$$

where $\langle \cdot, \cdot \rangle$ denotes the inner product and the $\log$ function is applied element wise. A lower SCE loss between the two predictions indicates better alignment. To assess overall long-short alignment, we compute the expectation over sequence lengths $l_1$ and $l_2$ for a given input $\mathbf{x}$:

$$\mathcal{L}_{\mathrm{misalign}}(g_\theta) = \mathbb{E}_{\mathbf{x}, l_1, l_2} \left[ \mathcal{L}_{\mathrm{SCE}}(g_\theta(\mathbf{x}_{[-l_1:]}), g_\theta(\mathbf{x}_{[-l_2:]})) \right].$$
$$(4)$$

We refer to this metric as the **long-short misalignment**, where a lower value signifies less discrepancy of output distributions across different lengths. An illustration of this metric is shown in Figure 2. In practice, we sample $l_1$ and $l_2$ from the interval $[l_{\mathrm{train}}/2, l_{\mathrm{train}}]$, where $l_{\mathrm{train}}$ represents the maximum context length used during training.

### 4.1.2. RESULTS

To evaluate the model's length generalization ability, we use the perplexity on long validation sets (16k length) and the LongBench-E score (Bai et al., 2023b). For perplexity evaluation, we select a subset from the RedPajama-Book

Table 1: The proposed long-short misalignment metric $\mathcal{L}_{\text{misalign}}$ of models, with their log of perplexity (PPL) on 16k-long contexts and LongBench-E score. We also provide $\mathcal{L}_{\text{train}}$ as an additional metric in comparison. We find that $\mathcal{L}_{\text{misalign}}$ correlates better with the long-context benchmark performance.

| Model | $\mathcal{L}_{\text{train}}$ | $\mathcal{L}_{\text{misalign}}$ | $\log(\text{PPL})$ | LongBench-E Score |
|---|---|---|---|---|
| GPT-J-6B (Wang, 2021) | 2.1 | 3.4 | 9.5 | 7.8 |
| GPT-NeoX-20B (Black et al., 2022) | 2.3 | 3.2 | 9.4 | 9.7 |
| Llama2-7B (Touvron et al., 2023) | 1.9 | 3.4 | 9.4 | 8.9 |
| RandomPos (Ruoss et al., 2023) | 2.0 | 2.8 | 8.2 | 9.2 |
| Yarn-Llama-2-7B-8k (Peng et al., 2024) | 2.0 | 2.6 | 3.8 | 21.2 |
| Qwen-7B-8k (Bai et al., 2023a) | 1.7 | 2.8 | 3.2 | 24.2 |
| CLEX-LLaMA-4K (Chen et al., 2024) | 1.9 | 2.4 | 1.8 | 32.7 |
| *Metric* | | | *Correlation Coefficient with Metric* | |
| $\mathcal{L}_{\text{train}}$ | | | 0.62 | -0.55 |
| $\mathcal{L}_{\text{misalign}}$ | | | 0.85 | -0.85 |

corpus (Computer, 2023), following the protocol in (Chen et al., 2024). LongBench-E is a multitask benchmark that comprehensively evaluates large language models' ability to understand long contexts, with task lengths averaging between 5k and 32k tokens.

**Empirical Results.** Table 1 shows the long-short misalignment metric $\mathcal{L}_{\text{misalign}}$ for the models along with their corresponding long-context evaluation results. Additionally, we include the training loss $\mathcal{L}_{\text{train}}$ for each model on the RedPajama-Book corpus, which reflects the perplexity on sequences with the maximum training length $l_{\text{train}}$. Interestingly, while the training loss (i.e., log of perplexity on sequences of length $l_{\text{train}}$) shows a moderate correlation with long-context performance metrics—indicating that lower training loss can contribute to improved length generalization—the long-short misalignment metric $\mathcal{L}_{\text{misalign}}$ demonstrates a much stronger correlation with long-context performance, as evidenced by its higher absolute correlation coefficient. These findings suggest that $\mathcal{L}_{\text{misalign}}$ is a promising indicator of length generalization ability. However, it is important to note that we do not claim any causal relationship based solely on these observations. We will elaborate more on this relationship in Section 5.

Additionally, we also provide theoretical support for this observation, extending previous work on autoregressive modeling (Zhang et al., 2024a) with a theorem:

**Theorem 4.1** (*Generalization guarantees for the natural language task*). *Under some model assumptions, the generalization error $\mathcal{E}_{\text{gen}}(g_\theta; l_{\text{test}})$ with testing length $l_{\text{test}}$ is upper bounded by the sum of training loss $\mathcal{L}_{\text{train}}(g_\theta)$ and misalignment metric $\mathcal{L}_{\text{misalign}}(g_\theta)$, i.e.,*

$$\mathcal{E}_{\text{gen}}(g_\theta^{l_{\text{train}}}; l_{\text{test}}) \leq C_1^{(l_{\text{test}})} \mathcal{L}_{\text{misalign}} + C_2^{(l_{\text{test}})} \mathcal{L}_{\text{train}} + C_0^{(l_{\text{test}})}, \tag{5}$$

*where $C_i^{(l_{\text{test}})} (i = 0, 1, 2)$ are constants related to $l_{\text{test}}$.*

*Specifically, each $C_i^{(l_{\text{test}})}$ increases as $l_{\text{test}}$ increases, and the ratio $C_1^{(l_{\text{test}})}/C_2^{(l_{\text{test}})}$ increases as $l_{\text{test}}$ increases. This indicates that as the testing length increases, the alignment loss becomes increasingly significant.*

**Proof Sketch.** We decompose the generalization error $\mathcal{E}_{\text{gen}}(g_\theta; l_{\text{test}})$ of long sequence into the misalignment term between this sequence and a set of shorter sequences, along with the prediction error of the shortest sequence. The prediction error of the shortest sequence is related to the model's training loss $\mathcal{L}_{\text{train}}$, while the misalignment term between long and short sequences increases significantly as the testing length grows. This is mainly because the model has a limited capacity to align sequences of different lengths, so aligning longer test sequences requires more intermediate-length sequences, resulting in higher alignment loss. Therefore, reducing the alignment loss can effectively lower the generalization error when testing with longer sequences. $\square$

The full statement and the proof are shown in Appendix C.2. This theorem highlights the importance of minimizing both $\mathcal{L}_{\text{misalign}}$ and $\mathcal{L}_{\text{train}}$ to achieve lower generalization error. Moreover, as the testing length $l_{\text{test}}$ increases, reducing $\mathcal{L}_{\text{misalign}}$ plays a more critical role in improving generalization performance. The above empirical and theoretical results motivate us to explicitly optimize the long-short misalignment metric to enhance the model's ability to handle long-context sequences, which will be stated in the following sections.

### 4.2. Long-Short Misalignment Metric as Regularization Term

Since both empirical and theoretical results indicate a strong correlation between the proposed long-short misalignment

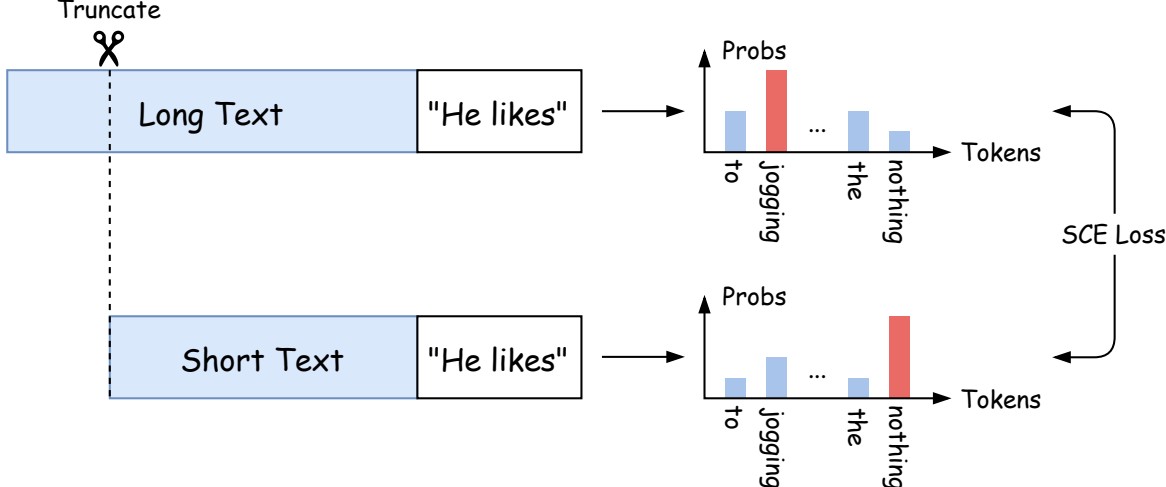

Figure 2: Illustration of long-short misalignment metric $\mathcal{L}_{\text{misalign}}$. Given two input sequences, where one is a truncated version of the other, the long-short misalignment metric is computed by taking the expectation on Symmetrical Cross-Entropy (SCE) loss (Wang et al., 2019) between the model's predictions for these two sequences.

metric and long-context benchmark performance, we incorporate this metric as a regularization term into the training loss, resulting in the new training loss defined as:

$$\mathcal{L}_{\text{train}}^*(g_\theta) = \mathcal{L}_{\text{train}}(g_\theta) + \alpha \cdot \mathcal{L}_{\text{misalign}}(g_\theta), \quad (6)$$

where $\mathcal{L}_{\text{train}}$ is the original cross-entropy training loss and $\alpha$ is the regularization coefficient. Calculating these two losses separately during training can be time-consuming, as the computation of $\mathcal{L}_{\text{misalign}}$ requires forward propagation through two distinct sequences. To address this, we propose an efficient implementation for $\mathcal{L}_{\text{train}}^*$. We first sample an integer $l_{\text{extra}}$ from $[1, l_{\text{train}}/2]$ and then sample a sequence of length $l_{\text{train}} + l_{\text{extra}}$. The first $l_{\text{train}}$ tokens form the first input sequence, while the last $l_{\text{train}}$ tokens form the second input sequence. Both sequences can be used to compute $\mathcal{L}_{\text{train}}$. The overlap between the two sequences starts at token $l_{\text{extra}} + 1$ and continues to token $l_{\text{train}}$, resulting in an overlap of $l_{\text{train}} - l_{\text{extra}}$ tokens. We can calculate the long-short misalignment loss in the overlapping positions. This implementation requires only two forward propagations for the two sequences, resulting in minimal additional time and resource costs compared to calculating the original train loss $\mathcal{L}_{\text{train}}$. A detailed Pytorch-like algorithm is provided in Appendix E and an overall illustration can be found in Figure 3. We will conduct experiments using the proposed regularization term in the next section to show its effectiveness.

## 5. Experiments on Natural Language Tasks

In this section, we conduct extensive experiments to verify the effectiveness of our proposed length alignment loss. We

Table 2: Performance of the fine-tuned models using only cross-entropy loss (baseline) and an additional long-short misalignment loss on long-context modeling benchmark, LongBench-E score (Bai et al., 2023b) and perplexity on the 8k-length validation set. The fine-tuning sequence length is 4k, exactly the same as the training sequence length. The models finetuned with our proposed loss outperform the baseline across different model adaption strategies.

| Benchmark | LongBench-E (↑) | | | Perplexity (↓) | | |
|---|---|---|---|---|---|---|
| Training steps | 50 | 100 | 200 | 50 | 100 | 200 |
| *RedPajama-Book* | | | | | | |
| $\mathcal{L}_{\text{train}}$ (Baseline) | 22.7 | 23.8 | 24.7 | 7.21 | 6.56 | 6.12 |
| $+0.1\mathcal{L}_{\text{misalign}}$ (Ours) | **23.1** | **25.2** | **26.6** | **6.89** | **6.24** | **5.88** |
| $+0.5\mathcal{L}_{\text{misalign}}$ (Ours) | 21.9 | 23.7 | 24.7 | 7.44 | 7.01 | 6.54 |
| *PG19* | | | | | | |
| $\mathcal{L}_{\text{train}}$ (Baseline) | 20.2 | 21.4 | 22.5 | **8.92** | **7.89** | 7.45 |
| $+0.1\mathcal{L}_{\text{misalign}}$ (Ours) | **20.7** | 22.1 | **25.3** | 8.95 | 7.92 | **7.35** |
| $+0.5\mathcal{L}_{\text{misalign}}$ (Ours) | 20.1 | **22.2** | 23.6 | 9.42 | 8.59 | 8.21 |

first examine our proposed training loss in Equation (6) on length generalization tasks, where the model is trained on short sequences and tested on longer ones. Additionally, we explore its application in another common scenario: long-context learning, where both training and testing involve long sequences. Then we conduct analytical experiments and ablation studies to further understand the impact of our proposed loss.

### 5.1. Experiments with Training on Short Sequences

In this section, we consider the classical length generalization setting, where the model is trained on short sequences

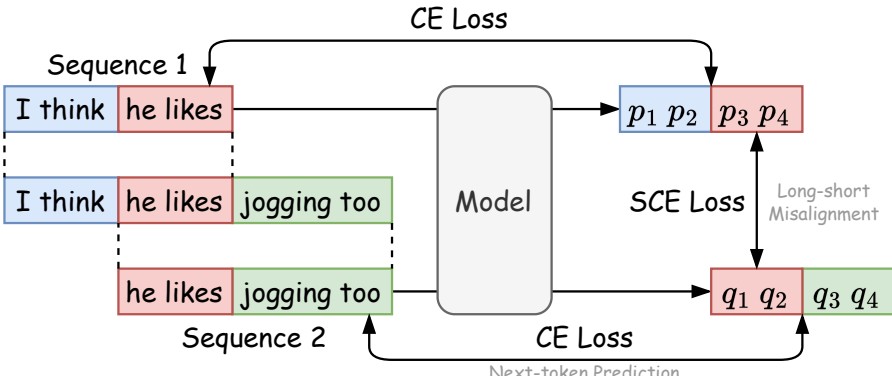

Figure 3: Illustration of efficiently calculating the total training loss $\mathcal{L}^*_{\text{train}}$. This implementation requires only two forward propagations for the two sequences, resulting in minimal additional time and resource costs

Table 3: Performance of the finetuned models using only cross-entropy loss (baseline) and an additional long-short misalignment loss on long-context modeling benchmark, LongBench-E score (Bai et al., 2023b) and perplexity on the 8k-length validation set. The fine-tuning sequence length is 8k. The models finetuned with our proposed loss outperform the baseline across different model adaption strategies.

| Benchmark | LongBench-E (↑) | | | Perplexity (↓) | | |
|---|---|---|---|---|---|---|
| Training steps | 50 | 100 | 200 | 50 | 100 | 200 |
| *LongQLora* | | | | | | |
| $\mathcal{L}_{\text{train}}$ (Baseline) | **21.9** | 22.1 | 23.4 | 6.82 | 6.41 | 5.82 |
| $+0.1\mathcal{L}_{\text{misalign}}$ (Ours) | 21.8 | 23.3 | **25.8** | **6.72** | **6.39** | **5.77** |
| $+0.5\mathcal{L}_{\text{misalign}}$ (Ours) | 21.4 | **23.9** | 25.1 | 7.07 | 6.62 | 5.92 |
| *EABF* | | | | | | |
| $\mathcal{L}_{\text{train}}$ (Baseline) | 22.1 | 22.9 | 23.6 | **6.89** | 6.52 | 6.01 |
| $+0.1\mathcal{L}_{\text{misalign}}$ (Ours) | **23.2** | **24.0** | **24.8** | 6.92 | **6.43** | **5.91** |
| $+0.5\mathcal{L}_{\text{misalign}}$ (Ours) | 22.5 | 23.2 | 23.9 | 7.14 | 6.78 | 6.34 |

(4k-long) and tested on longer sequences (at least 5k-long). Due to the high computational cost of pre-training large language models from scratch, most current methods fine-tune open-sourced pre-trained models (Chen et al., 2024; Yang, 2023; Peng et al., 2024). In our experiments, we use Llama2-7b (Touvron et al., 2023) as the base model and apply the CLEX (Chen et al., 2024) adjustment method. We use two datasets: the RedPajama-Book corpus (Computer, 2023) and PG19 (Rae et al., 2019). The experiments are conducted with a context length of 4,096, a batch size of 64, and a maximum of 200 training steps. For the regularization coefficient $\alpha$, we test values of 0.1 and 0.5.

We evaluate performance using the LongBench-E score (Bai et al., 2023b) and perplexity on validation sets made up of sequences of length 8,192 from the corpus of the respective training dataset. LongBench-E is a multitask benchmark that comprehensively evaluates large language models' abil-ity to understand long contexts, with task lengths averaging between 5k and 32k tokens, which has been adopted by many previous works (Chen et al., 2024; Jin et al., 2024) as an effective evaluation metric for long-context modeling. The results, shown in Table 2, indicate that the model fine-tuned with our proposed loss consistently outperforms the baseline model on the LongBench-E benchmark. The fine-tuned model shows lower perplexity on RedPajama-Book and similar perplexity on PG19. These results support the effectiveness of our proposed loss and the intuition that lower misalignment metric $\mathcal{L}_{\text{misalign}}$ leads to better length generalization ability. For the regularization coefficient $\alpha$, we find that larger values do not always improve performance, as they may interfere with the model's next-word prediction.

### 5.2. Experiments with Training on Longer Sequences

In this section, we consider the scenario that the model is finetuned on a longer sequence than the training length. We use different model adjustment strategies during the fine-tuning stage, to demonstrate that the proposed length alignment loss can be applied to various long-context fine-tuning methods. Our experiments use Llama2-7b as the base model. For model adjustments, we consider two approaches: LongQLora (Yang, 2023) and EABF (Zhang et al., 2024b). LongQLora leverages multiple techniques, including Position Interpolation (Chen et al., 2023a), QLoRA (Dettmers et al., 2024), and Shift Short Attention from LongLoRA (Chen et al., 2023b). Meanwhile, EABF introduces a dynamic rescaling mechanism to the attention layers and applies a higher base frequency for RoPE. The experiments are conducted on the RedPajama-Book corpus (Computer, 2023), with a context length of 8,192, a batch size of 64, and a maximum of 200 training steps. For the regularization coefficient $\alpha$, we test values of 0.1 and 0.5.

Table 4: The overall evaluation results on BABILong (Kuratov et al., 2024) with sequence lengths of 4K, 8K, 16K.

| Training Loss | Evaluation Length | | |
|---|---|---|---|
| | 4K | 8K | 16K |
| $\mathcal{L}_{\text{train}}$ (Baseline) | 48.2 | 42.4 | 37.9 |
| $\mathcal{L}_{\text{train}} + 0.1\mathcal{L}_{\text{misalign}}$ | **49.1** | **44.4** | **40.1** |

Table 5: The evaluation results on BABILong with different locations of the facts in the QA1 task. Input length is 16K.

| Training Loss | Fact Depth (%) | | | |
|---|---|---|---|---|
| | 0 | 25 | 50 | 75 |
| $\mathcal{L}_{\text{train}}$ (Baseline) | **75** | 64 | 30 | 69 |
| $\mathcal{L}_{\text{train}} + 0.1\mathcal{L}_{\text{misalign}}$ | 73 | 64 | **38** | **74** |

We evaluate performance using the LongBench-E score and perplexity on validation sets composed of sequences of length 8,192 from the RedPajama-Book corpus. The results are shown in Table 3. Notably, our methods outperform the baseline across both model adjustment strategies. Specifically, models trained with our proposed regularization term achieve up to a $2.4\%$ improvement in the LongBench-E score. Similar to the previous experiments, we observe that excessively large regularization coefficient values may not consistently benefit long-context modeling, which is reflected in the slightly lower LongBench-E scores and higher perplexity, indicating that overly strong regularization may disrupt the model's training process.

### 5.3. Experiments on BABILong

To further assess the ability to retrieve and utilize distant information, we conduct extensive experiments on BABILong (Kuratov et al., 2024), a challenging reasoning-in-a-haystack task specifically designed for evaluating long-context capabilities. BABILong comprises question-answering tasks in which the supporting facts for each question are situated at specific positions within the context. Using the model setup described in Section 5.1, which incorporates CLEX as the adjustment method and RedPajama-Book as the training dataset, all models are fine-tuned for 200 steps. The evaluation is performed on input sequences of lengths 4K, 8K, and 16K, with the overall results summarized in Table 4. The results indicate that our proposed method consistently outperforms the baseline across all evaluated lengths. Specifically, our method achieves a performance gain of $2.0\%$ at length 8K and $2.2\%$ at length 16K.

Additionally, we analyze the impact of the supporting fact's position within the input context using the BABILong QA1 task, where each question is associated with a single supporting fact. The results of this analysis are presented in Table 5, offering two key insights: (1) **Performance with**

Table 6: Ablation study on the regularization coefficient $\alpha$. The setting is the same as Table 2. We adopt RedPajama-Book (Computer, 2023) as the training dataset. We find it important to select a moderate value for $\alpha$.

| Benchmark | LongBench-E ($\uparrow$) | | | Perplexity ($\downarrow$) | | |
|---|---|---|---|---|---|---|
| Training steps | 50 | 100 | 200 | 50 | 100 | 200 |
| *RedPajama-Book* | | | | | | |
| $\mathcal{L}_{\text{train}}$ (Baseline) | 22.7 | 23.8 | 24.7 | 7.21 | 6.56 | 6.12 |
| $+0.1\mathcal{L}_{\text{misalign}}$ | 23.1 | 25.2 | 26.6 | **6.89** | **6.24** | **5.88** |
| $+0.3\mathcal{L}_{\text{misalign}}$ | **23.4** | **25.8** | **27.1** | 6.95 | 6.35 | 5.98 |
| $+0.5\mathcal{L}_{\text{misalign}}$ | 21.9 | 23.7 | 24.7 | 7.44 | 7.01 | 6.54 |
| $+1.0\mathcal{L}_{\text{misalign}}$ | 18.2 | 19.4 | 19.9 | 16.21 | 14.12 | 12.92 |

Table 7: Ablation study on the sampling range. The setting is the same as Table 3. We adopt RedPajama-Book (Computer, 2023) as the training datasets and LongQLora (Yang, 2023) as the model adjustment method. We find it important to carefully balance the sampling range to optimize the model's generalization to longer contexts.

| Benchmark | LongBench-E ($\uparrow$) | | | Perplexity ($\downarrow$) | | |
|---|---|---|---|---|---|---|
| Training steps | 50 | 100 | 200 | 50 | 100 | 200 |
| Sampling range of $l_{\text{extra}}$ | | | | | | |
| (1) $[1, l_{\text{train}}/2]$ (Current) | **21.8** | **23.3** | **25.8** | **6.72** | 6.39 | **5.77** |
| (2) $[1, l_{\text{train}}/4]$ | 21.5 | 23.2 | 25.7 | 6.77 | **6.29** | 5.81 |
| (3) $[l_{\text{train}}/4, l_{\text{train}}/2]$ | 21.4 | 22.7 | 24.5 | 6.82 | 6.47 | 5.94 |
| (4) $[1, l_{\text{train}}]$ | 18.2 | 18.9 | 19.1 | 15.65 | 13.59 | 12.52 |

**early-context facts:** When the supporting fact is located at the beginning of the input context (fact depth = 0), our method achieves performance comparable to the baseline. This suggests that despite the form of the regularization potentially encouraging the model to neglect earlier contexts, it does not lead to this behavior in practice. (2) **Performance with middle-context facts:** When the supporting fact is positioned in the middle of the context (fact depth = 50 or 75), our method shows considerable improvement over the baseline. This indicates that our approach effectively mitigates the "loss-in-the-middle" phenomenon (Liu et al., 2024), a common challenge in large language models. Together, these results strongly support the effectiveness of our proposed regularization term in enhancing length generalization ability, particularly for tasks requiring attention across diverse positions.

### 5.4. Ablation Studies

Since we incorporate several hyperparameters such as the regularization coefficient $\alpha$ and the sampling range $|l_1 - l_2|$ in the misalignment metric, we conduct extensive experiments to explore how these hyperparameters affect the model performance.

**Regularization coefficient $\alpha$.** In addition to the settings already provided in the previous experiments ($\alpha = 0$ as the baseline, $\alpha = 0.1$, and $\alpha = 0.5$), we evaluated $\alpha = 0.3$

Table 8: Comparison between synthetic tasks and natural language tasks.

| | Synthetic Tasks | Language Tasks |
|---|---|---|
| Output Space | $\mathbb{R}$ | $L_1$ unit ball in $\mathbb{R}^{|\mathcal{V}|}$ |
| Specific Task | Length/Sum prediction | Next token prediction |
| Long-Short Misalignment | Exist | Exist |
| Priori on Output Distribution | Explicit and predifined | Implicit and task-dependent |
| Alignment Technique | Explicit reparameterization | Regularization term across lengths |
| Does the technique decrease long-short misalignment? | Yes (explicitly) | Yes (implicitly through optimization) |
| Does the technique improve length generalization? | Yes | Yes |

and $\alpha = 1.0$ under the same experimental conditions as Table 2, using CLEX as the model adjustment. We still adopt RedPajama-Book as the training dataset. The results are shown in Table 6, which reveal the following trend: (1) Performance peaks for $\alpha$ values in the range $[0.1, 0.3]$ in both evaluation metrics. (2) Larger values of $\alpha$ (e.g., $\alpha = 0.5$ or $\alpha = 1.0$) lead to a significant decline in performance, confirming the risks of over-regularization. These findings highlight the importance of selecting a moderate value for $\alpha$. We suggest using a coefficient $\alpha$ between 0.1 and 0.3 as default to mitigate the risk of over-regularization.

**Sampling range.** In equation 4, we sample $l_1$ and $l_2$ from $[l_{\text{train}}/2, l_{\text{train}}]$ by default to avoid input sequence with significantly different lengths. This is equivalent to sampling $l_{\text{extra}}$ from $[1, l_{\text{train}}/2]$. Here we conduct an ablation study examining how different sampling strategies of $l_{\text{extra}}$ affect performance. We consider four sampling configurations: (1) The current strategy, sampling from $[1, l_{\text{train}}/2]$; (2) Sampling from a narrower range $[1, l_{\text{train}}/4]$; (3) Sampling from a narrower range $[l_{\text{train}}/4, l_{\text{train}}/2]$; (4) Sampling from a broader range $[1, l_{\text{train}}]$ and remove the limit that $l_1$ and $l_2$ should be in $[l_{\text{train}}/2, l_{\text{train}}]$. We conduct experiments using the same setting as Table 3, using LongQLora for model adjustments and a regularization coefficient of 0.1. The results are shown in Table 7: (1) Setting 2 achieved performance comparable to the current strategy, while Setting 3 showed slightly inferior performance compared to the current strategy. This suggests that **aligning outputs between sequences with moderate length discrepancies effectively supports long-context modeling.** (2) Setting 4 yielded significantly worse performance than the current strategy, indicating that **encouraging alignment between sequences with large length differences adversely affects the model's long-context capabilities.** These results underscore the importance of carefully balancing the sampling range in the proposed regularization.

We also compare our methods with baselines under the same computational time in Appendix F.

## 6. Discussion

Since our work is initially motivated by phenomena observed in synthetic tasks, we provide additional clarification on the relationship between synthetic tasks and natural language tasks by summarizing their key differences and similarities in Table 8. While these two types of tasks differ significantly in their specific forms and output space, they share a common challenge: **output distribution misalignment across different input lengths**. Our analysis highlights that employing an alignment technique–whether explicit reparameterization in synthetic tasks or regularization in natural language tasks–can effectively mitigate this misalignment. This mitigation directly enhances the model's length generalization ability, demonstrating the broader applicability of our approach.

## 7. Conclusion

In this work, we investigated the challenges of long-context modeling in large language models and introduced a fresh perspective by examining the output behavior of models across varying input lengths. We identified that **misalignment in output distributions across sequences of different lengths**, which we term *long-short misalignment*, plays a critical role in limiting length generalization—a classical and foundational subproblem in long-context modeling. Through both synthetic and natural language tasks, we demonstrated that long-short misalignment is strongly correlated with performance on long-context inputs. We further proposed a regularization term based on this metric to explicitly reduce output divergence during training. Extensive experiments confirm that this approach not only improves length generalization but also leads to more effective long-context modeling. Overall, our work highlights the importance of considering the output space when designing and analyzing models for long-context scenarios, offering a new dimension for future research in scalable and effective language modeling.

## Acknowledgements

Yisen Wang was supported by National Key R&D Program of China (2022ZD0160300), National Natural Science Foundation of China (92370129, 62376010), and Beijing Nova Program (20230484344, 20240484642).

## Impact Statement

We believe there are no direct ethical problems in this work since it primarily focuses on the theoretical analysis and improvement of large language models (LLMs). However, LLMs can generate inaccurate or harmful content, and this research does not offer a direct solution to these issues. Users are encouraged to ensure that the LLMs obey the ethical standards when implementing the proposed methods.

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

# A. Full Related Work

**Length Generalization on Synthetic Tasks.** Our paper is related to the line of work that seeks to understand the capabilities and limitations of Transformer models when it comes to algorithmic reasoning (Veličković & Blundell, 2021). Specifically, we focus on simple tasks and study length generalization on the standard Transformer architecture with causal structure. Related to this, Lee et al. (2023) study how well transformers trained from scratch can learn simple arithmetic tasks, and find that no length generalization is observed. Nogueira et al. (2021) find that partial length generalization on addition is observed only when models reach 3B parameters and when the addition questions are presented in reverse order. Jelassi et al. (2023) study models trained on addition and find strong generalization performance when using a few examples of longer sequences. Zhou et al. (2024) proposes the RASP Generalization Conjecture that Transformers tend to learn a length-generalizing solution if there exists a short RASP-L program that works for all input lengths. Liu et al. (2023) discovers that the Transformer will learn shortcuts through the study of various synthetic tasks. Besides these explorations on specific tasks, some works study the impact of different positional encodings on the length generalization of math reasoning tasks. Alibi adds a linear bias on the attention score to achieve better length generalization performance (Press et al., 2021). Ontanon et al. (2022) studies different settings of positional encodings and identifies Transformer configurations that generalize compositionally significantly better in a diverse set of compositional tasks. Kazemnejad et al. (2023) systematically studies the role of no positional encoding (NoPE) in Transformer with causal structure. Ruoss et al. (2023) proposes a randomized positional encoding scheme that simulates the positions of longer sequences and randomly selects an ordered subset to fit the sequence's length.

**Long-context Modeling on Natural Language Tasks.** A series of works (Sun et al., 2023; Chi et al., 2022; 2023; Zhang et al., 2024b; Chen et al., 2024; Peng et al., 2024; Yang, 2023; Chen et al., 2023a) aim to extend the context size of Transformer-based models during fine-tuning, primarily by modifying positional encodings. For example, Zhang et al. (2024b) introduces a novel extension to RoPE (Su et al., 2024) which combines adjusting RoPE's base frequency and scaling the attention logits to help LLMs efficiently adapt to a larger context window. Chen et al. (2024) generalizes the positional encoding scaling approaches to model the continuous dynamics by ordinary differential equations over the length scaling factor. (Chen et al., 2023a) proposes to extend the context length by slightly modifying RoPE via Position Interpolation (PI) and fine-tuning on a small amount of data. Our approach, in contrast, focuses on the model's output space, which identifies the crucial role of long-short alignment in length generalization. Wang et al. (2024a) proposes a novel approach designed to narrow the generalization gap by refining the interpolation of RoPE features for OOD positions and provides length extrapolation analysis on the feature gap.

# B. Model Details for Synthetic Tasks

We focus on the decoder-only Transformer (Vaswani et al., 2017), a model widely used in both synthetic tasks (Zhou et al., 2024; Jelassi et al., 2023) and LLMs (Touvron et al., 2023; Peng et al., 2024) which utilizes a causal mask in the self-attention module to enable auto-regressive generation. We consider several positional encodings: learnable positional encoding (Radford et al., 2019), Alibi (Press et al., 2021), rotary positional encoding (Su et al., 2024) and no positional encoding (NoPE). Since recent works found that by removing the positional encoding, Transformers can trained to be well generalized on length (Deletang et al., 2022; Kazemnejad et al., 2023), we adopt this setting (i.e. NoPE) by default. To provide a clear signal for the model to accomplish the tasks, we add both the begin-of-sentence (BOS) token and the end-of-sentence (EOS) token in the sequence. We apply a feed-forward neural network on the hidden state of the last token to generate an output of a real number. We train all of our models on the train distribution from scratch to convergence if possible. For all tasks, the length of training examples is sampled uniformly from length 1 up to the max training length $l_{\text{train}}$. We select hyper-parameters such as the learning rate for each task based on what is required to fit the training set. At test time, the length of the examples will traverse from 1 to the max testing length $l_{\text{test}}$.

# C. Theorems and Proofs

### C.1. Theoretical Analysis for Synthetic Tasks

Ahn et al. (2024) suggests that linear Transformers serve as realistic abstractions for understanding Transformer optimization and generalization. Therefore, following (Ahn et al., 2024; Zhang et al., 2024a), our analysis is based on the linear attention model. The general form of linear attention is given by:

$$\text{Attn}(\mathbf{x}) = QK^{\top}V = \mathbf{x}W^{Q}(\mathbf{x}W^{K})^{\top}\mathbf{x}W^{V}, \tag{7}$$

where $W^Q, W^K, W^V$ are projections, and $n$ is the length of input $\mathbf{x}$. In all tasks, we will use a linear attention model with a causal mask. Specifically, we normalize the output according to its position, which allows linear attention to perform similarly to dot-product attention. For example, the $k$-th output will be normalized as follows:

$$\frac{1}{k}Q_k K^\top V = \frac{1}{k}\sum_{i=1}^{k} Q_k K_i^\top V_i. \tag{8}$$

For the synthetic tasks, we add bias terms to $Q, K, V$ to mitigate the impact of 0 in the input. The model based on linear attention is defined as follows:

$$g_\theta(\mathbf{x}_{[:k]}) = \frac{1}{k}\sum_{i=1}^{k} Q_k K_i^\top V_i, \tag{9}$$

and it is trained based on the following target function:

$$\mathcal{L}(g_\theta; l_{\text{train}}) = \mathbb{E}_{\mathbf{x}\in\{0,1\}^{l_{\text{train}}}}\left[\frac{1}{l_{\text{train}}}\sum_{i=1}^{l_{\text{train}}}\left\|g_\theta(\mathbf{x}_{[:i]}) - y(\mathbf{x}_{[:i]})\right\|_2^2\right], \tag{10}$$

where $\mathbf{x}_{[:i]}$ indicates the first $i$-tokens of input $\mathbf{x}$. The optimal model trained on sequence with maximum length of $l_{\text{train}}$ is denoted as $g_\theta^{l_{\text{train}}}$. We have the following result:

**Theorem C.1.** *In the length prediction task and the sum prediction, the length generalization loss has a quadratic relationship with the predicted length, i.e.,*

$$\mathcal{E}_{\text{length}}(g_\theta^{l_{\text{train}}}; l_{\text{test}}) = \mathbb{E}_{\mathbf{x}_{\text{test}}\in\{0,1\}^{l_{\text{test}}}}\left[\left\|g_\theta^{l_{\text{train}}}(\mathbf{x}_{\text{test}}) - y(\mathbf{x}_{\text{test}})\right\|_2^2\right] = \mathcal{O}\left((l_{\text{test}} - l_{\text{train}})^2\right), \tag{11}$$

$$\mathcal{E}_{\text{sum}}(g_\theta^{l_{\text{train}}}; l_{\text{test}}) = \mathcal{O}\left((l_{\text{test}} - l_{\text{train}})^2\right). \tag{12}$$

*However, in the mean prediction task, the length generalization loss has a fixed upper bound:*

$$\mathcal{E}_{\text{mean}}(g_\theta^{l_{\text{train}}}; l_{\text{test}}) = \mathcal{O}(1). \tag{13}$$

*Proof.* Since the input consists of 0 and 1, the $Q_k, K_k, V_k$ will each have only two possible values. Thus, we may assume that when the input token is 1, the $Q_k, K_k, V_k$ are $q', k', v'$ respectively, and when the input token is 0, the $Q_k, K_k, V_k$ are $q'', k'', v''$ respectively. Furthermore, we note that in (9), $K_i$ and $V_i$ always share the same subscript. Therefore, we can treat them as a single token and replace them with $\kappa$, where we define $\kappa' = k'v'$ and $\kappa'' = k''v''$. We can decompose (10) into $l_{\text{train}}$ separate part:

$$\mathcal{L}(g_\theta; l_{\text{train}}) = \frac{1}{l_{\text{train}}}\sum_{l=1}^{l_{\text{train}}} \ell(g_\theta; l), \quad \ell(g_\theta; l) = \mathbb{E}_{\mathbf{x}}\left[\left\|g_\theta(\mathbf{x}_{[:l]}) - y(\mathbf{x}_{[:l]})\right\|_2^2\right]. \tag{14}$$

Next, we consider each task individually.

(a) First, we study the length prediction task. In this case, $y(\mathbf{x}_{[:l]}) = l$. Expanding $\ell(g_\theta; l)$ we have:

$$\ell(g_\theta; l) = \mathbb{E}_{\mathbf{x}}\left[\left\|g_\theta(\mathbf{x}_{[:l]}) - y(\mathbf{x}_{[:l]})\right\|_2^2\right]$$
$$= \frac{1}{2^l}\sum_{i=0}^{l} C_{l-1}^{i-1}\left(\frac{1}{l}(iq'\kappa' + (l-i)q'\kappa'') - l\right)^2 + C_{l-1}^{i}\left(\frac{1}{l}(iq''\kappa' + (l-i)q''\kappa'') - l\right)^2.$$

It's easy to find that $\ell(g_\theta, l)$ achieves its minimum 0 if and only if $q', q'', \kappa', \kappa''$ satisfy the following conditions:

$$\begin{cases} q' = q'' \neq 0, \kappa' = \kappa'' \neq 0, \\ q'\kappa' = q''\kappa' = q'\kappa'' = q''\kappa'' = l. \end{cases} \tag{15}$$

These are also properties that the solution of $\partial\ell(g_\theta; l) = 0$ holds. Note that the first property holds for any $\ell(g_\theta; l)$ and is independent of $l$. Since $\mathcal{L}(g_\theta; l_{\text{train}})$ is composed of $\ell(g_\theta; l)$, and each optimal solution of $\ell(g_\theta; l)$ satisfying $q' = q'' \neq 0, \kappa' = \kappa'' \neq 0$, then the global solution for $\partial\mathcal{L}(g_\theta; l_{\text{train}}) = 0$ should also have this property. Therefore, we may assume that $q'\kappa' = q''\kappa' = q'\kappa'' = q''\kappa'' = \Gamma$, and our goal is to find the $\Gamma$ that satisfying $\partial\mathcal{L}(g_\theta; l_{\text{train}})/\partial\Gamma = 0$, which means that this $\Gamma$ minimizes $\mathcal{L}(g_\theta; l_{\text{train}})$. In this case, it is easy to find that $\ell(g_\theta; l) = (\Gamma - l)^2$, which means that $\partial\ell(g_\theta; l)/\partial\Gamma = 2(\Gamma - l)$. Therefore, we have:

$$\frac{\partial\mathcal{L}(g_\theta; l_{\text{train}})}{\partial\Gamma} = \frac{2}{l_{\text{train}}} \sum_{l=1}^{l_{\text{train}}} (\Gamma - l), \tag{16}$$

solving $\partial\mathcal{L}(g_\theta; l_{\text{train}})/\partial\Gamma = 0$ and we have $\Gamma = (l_{\text{train}} + 1)/2$, so the $g_\theta^{l_{\text{train}}}$ satisfying $q'\kappa' = q''\kappa' = q'\kappa'' = q''\kappa'' = (l_{\text{train}} + 1)/2$. Therefore, we have:

$$\mathcal{E}_{\text{length}}(g_\theta^{l_{\text{train}}}; l_{\text{test}}) = \ell(g_\theta^{l_{\text{train}}}; l_{\text{test}}) = \left(l_{\text{test}} - \frac{l_{\text{train}} + 1}{2}\right)^2 = \mathcal{O}\left((l_{\text{test}} - l_{\text{train}})^2\right). \tag{17}$$

(b) We now study the sum prediction task. In this case, $y(\mathbf{x}_{[:l]}) = \sum_{i=1}^{l} \mathbf{x}_i$. Expanding $\ell(g_\theta; l)$ we have:

$$\ell(g_\theta; l) = \mathbb{E}_{\mathbf{x}} \left[\left\|g_\theta(\mathbf{x}_{[:l]}) - y(\mathbf{x}_{[:l]})\right\|_2^2\right]$$

$$= \frac{1}{2^l} \sum_{i=0}^{l} C_{l-1}^{i-1} \left(\frac{1}{l}(iq'\kappa' + (l-i)q'\kappa'') - i\right)^2 + C_{l-1}^{i} \left(\frac{1}{l}(iq''\kappa' + (l-i)q''\kappa'') - i\right)^2.$$

It's easy to find that $\ell(g_\theta, l)$ achieves its minimum 0 if and only if $q', q'', \kappa', \kappa''$ satisfy the following conditions:

$$\begin{cases} q' = q'' \neq 0, \kappa' \neq 0, \kappa'' = 0, \\ q'\kappa' = q''\kappa' = l. \end{cases} \tag{18}$$

This is similar to the previous conditions (15). Thus, we can perform a similar analysis as above, and we similarly assume that $q'\kappa' = q''\kappa' = \Gamma$ with $\kappa'' = 0$. In this case, we have:

$$\ell(g_\theta; l) = \frac{1}{2^l l^2} \sum_{i=0}^{l} i^2 C_l^i (\Gamma - l)^2 = \frac{l+1}{2l}(\Gamma - l)^2. \tag{19}$$

Thus we have:

$$\frac{\partial\ell(g_\theta; l)}{\partial\Gamma} = \frac{l+1}{l}(\Gamma - l), \tag{20}$$

which leads to:

$$\frac{\partial\mathcal{L}(g_\theta; l_{\text{train}})}{\partial\Gamma} = \frac{1}{l_{\text{train}}} \sum_{l=1}^{l_{\text{train}}} \frac{\partial\ell(g_\theta; l)}{\partial\Gamma} = \frac{1}{l_{\text{train}}} = (l_{\text{train}} + H_{l_{\text{train}}})\Gamma - \frac{l_{\text{train}}(l_{\text{train}} + 3)}{2}, \tag{21}$$

where $H_n$ is the $n$-th harmonic number, i.e., $H_n = \sum_{i=1}^{n} 1/i$. Solving $\partial\mathcal{L}(g_\theta; l_{\text{train}})/\partial\Gamma = 0$ and we have $\Gamma = l_{\text{train}}(l_{\text{train}} + 3)/2(l_{\text{train}} + H_{l_{\text{train}}})$, so the $g_\theta^{l_{\text{train}}}$ satisfying $q'\kappa' = q''\kappa' = l_{\text{train}}(l_{\text{train}} + 3)/2(l_{\text{train}} + H_{l_{\text{train}}})$. Therefore, we have:

$$\mathcal{E}_{\text{sum}}(g_\theta^{l_{\text{train}}}; l_{\text{test}}) = \ell(g_\theta^{l_{\text{train}}}; l_{\text{test}}) = \frac{l_{\text{test}} + 1}{2l_{\text{test}}} \left(\frac{l_{\text{train}}(l_{\text{train}} + 3)}{2(l_{\text{train}} + H_{l_{\text{train}}})} - l_{\text{test}}\right)^2$$

$$\approx \frac{1}{2} \left(\frac{l_{\text{train}} + 3}{2 + \text{const}} - l_{\text{test}}\right)^2 \tag{22}$$

$$= \mathcal{O}\left((l_{\text{test}} - l_{\text{train}})^2\right).$$

(c) Finally we study the mean prediction task. In this case, $y(\mathbf{x}_{[:l]}) = \sum_{i=1}^{l} \mathbf{x}_i / l$. Expanding $\ell(g_\theta; l)$ we have:

$$\ell(g_\theta; l) = \mathbb{E}_{\mathbf{x}} \left[ \left\| g_\theta(\mathbf{x}_{[:l]}) - y(\mathbf{x}_{[:l]}) \right\|_2^2 \right]$$

$$= \frac{1}{2^l} \sum_{i=0}^{l} C_{l-1}^{i-1} \left( \frac{1}{l}(iq'\kappa' + (l-i)q'\kappa'') - \frac{i}{l} \right)^2 + C_{l-1}^i \left( \frac{1}{l}(iq''\kappa' + (l-i)q''\kappa'') - \frac{i}{l} \right)^2.$$

It's easy to find that $\ell(g_\theta, l)$ achieves its minimum 0 if and only if $q', q'', \kappa', \kappa''$ satisfy the following conditions:

$$\begin{cases} q' = q'' \neq 0, \kappa' \neq 0, {\color{red}\kappa'' = 0}, \\ q'\kappa' = q''\kappa' = 1. \end{cases} \tag{23}$$

This is similar to the previous conditions (18). In fact, the above properties are irrelevant to $l$, so for all $\ell(g_\theta, l)$ these properties hold. Therefore, in this case, the optimal solution of $\mathcal{L}(g_\theta; l_{\text{train}})$ will satisfy (23), which leads to $\mathcal{L}(g_\theta; l_{\text{train}}) = 0$. Under this situation, the length generalization error is:

$$\mathcal{E}_{\text{sum}}(g_\theta^{l_{\text{train}}}; l_{\text{test}}) = \ell(g_\theta^{l_{\text{train}}}; l_{\text{test}}) = 0. \tag{24}$$

Consider the case where the solution is not optimal, i.e., when $q'\kappa' = q''\kappa'' = 1 + \varepsilon$, where $\varepsilon \neq 0$ is small, we can similarly obtain:

$$\mathcal{E}_{\text{sum}}(g_\theta^{l_{\text{train}}}; l_{\text{test}}) = \ell(g_\theta^{l_{\text{train}}}; l_{\text{test}}) = \frac{(l_{\text{test}} + 1)\varepsilon^2}{2 l_{\text{test}}} \leq \varepsilon^2. \tag{25}$$

In conclusion, we have:

$$\mathcal{E}_{\text{sum}}(g_\theta^{l_{\text{train}}}; l_{\text{test}}) = \mathcal{O}(1), \tag{26}$$

which completes the proof.

$\square$

## C.2. Theoretical Analysis for Natural Language Tasks

We now focus on natural language tasks. We make some changes to the model following (Zhang et al., 2024a). In natural language tasks, the model requires an additional projection $W$ to make the output a probability distribution. That is, the model is modified as follows:

$$g_\theta(\mathbf{x}_{[:k]}) = \frac{1}{k} \sum_{i=1}^{k} Q_k K_i^\top V_i W, \tag{27}$$

In this case, each token $\mathbf{x}_i$, output $g_\theta(\mathbf{x})$ and the objective function $y(\mathbf{x})$ are normalized. Without loss of generality, we may assume that the changes in the objective function after truncating the inputs are negligible, i.e.

$$\mathbf{Pr}(y(\mathbf{x}_{[:l_1]}) \neq y(\mathbf{x}_{[:l_2]})) = 0 \quad (\forall l_1, l_2). \tag{28}$$

For simplicity, we use the $L_2$-norm instead of SCE to measure misalignment, i.e.:

$$\mathcal{L}_{\text{misalign}}(g_\theta) = \mathbb{E}_{\mathbf{x}, l_1, l_2} \left[ \| g_\theta(\mathbf{x}_{[-l_1:]}) - g_\theta(\mathbf{x}_{[-l_2:]}) \|_2^2 \right], \tag{29}$$

and we use the $L_2$-norm instead of the CE loss as the training loss function since these two functions differ only by a constant when the output and target are regularized. Under these conditions, we have the following result:

**Theorem C.2 (*Generalization guarantees for the natural language task*).** *Suppose that $g_\theta^{l_{\text{train}}}$ is the model trained on sequences with maximum training length $l_{\text{train}}$. When the testing length $l_{\text{test}}$ satisfying $l_{\text{test}} > l_{\text{train}}$, the generalization loss $\mathcal{E}_{\text{gen}}(g_\theta^{l_{\text{train}}}; l_{\text{test}})$ has the following upper bound:*

$$\mathcal{E}_{\text{gen}}(g_\theta^{l_{\text{train}}}; l_{\text{test}}) \leq C_1^{(l_{\text{test}})} \cdot \mathcal{L}_{\text{misalign}}(g_\theta^{l_{\text{train}}}) + C_2^{(l_{\text{test}})} \cdot \mathcal{L}_{\text{train}}(g_\theta^{l_{\text{train}}}) + C_0^{(l_{\text{test}})}, \tag{30}$$

*where $C_i^{(l_{\text{test}})}(i = 0, 1, 2)$ are constants related to $l_{\text{test}}$. Specifically, the $C_i^{(l_{\text{test}})}$ increase as the $l_{\text{test}}$ increases and the ratio $C_1^{(l_{\text{test}})} / C_2^{(l_{\text{test}})}$ increases as $l_{\text{test}}$ increases. This indicates that as the testing length increases, the alignment loss becomes increasingly significant.*

*Proof.* Suppose that

$$l_{\text{extra}} = \begin{cases} \dfrac{l_{\text{train}}}{2} - l, & l < \dfrac{l_{\text{train}}}{2}, \\[2ex] l_{\text{train}} - l, & l \geq \dfrac{l_{\text{train}}}{2}. \end{cases} \tag{31}$$

where $l \in [1, l_{\text{train}}]$ is an arbitrary integer. Let $l^{(k)} = l_{\text{train}} + k \cdot l_{\text{extra}}, (k = 0, 1, 2, \cdots, N_l)$, where $N_l = \lfloor (l_{\text{test}} - l_{\text{train}})/l_{\text{extra}} \rfloor$, we have:

$$\begin{aligned} \left\| g_\theta^{l_{\text{train}}}(\mathbf{x}_{[-l_{\text{test}}:]}) - y(\mathbf{x}_{[-l_{\text{test}}:]}) \right\|_2^2 \leq (N_l + 2) & \left( \left\| g_\theta^{l_{\text{train}}}(\mathbf{x}_{[-l_{\text{test}}:]}) - g_\theta^{l_{\text{train}}}(\mathbf{x}_{[-l^{(N_l)}:]}) \right\|_2^2 \right. \\ & + \sum_{k=0}^{N_l-1} \left\| g_\theta^{l_{\text{train}}}(\mathbf{x}_{[-l^{(k+1)}:]}) - g_\theta^{l_{\text{train}}}(\mathbf{x}_{[-l^{(k)}:]}) \right\|_2^2 \\ & \left. + \left\| g_\theta^{l_{\text{train}}}(\mathbf{x}_{[-l_{\text{train}}:]}) - y(\mathbf{x}_{[-l_{\text{test}}:]}) \right\|_2^2 \right). \end{aligned} \tag{32}$$

For each $k \in [0, N_l - 1]$, we have:

$$\begin{aligned} & \left\| g_\theta^{l_{\text{train}}}(\mathbf{x}_{[-l^{(k+1)}:]}) - g_\theta^{l_{\text{train}}}(\mathbf{x}_{[-l^{(k)}:]}) \right\|_2^2 \\ &= \left\| \frac{1}{l^{(k+1)}} \mathbf{x}_0 W^Q (W^K)^\top \mathbf{x}_{[-l^{(k+1)}:]}^\top \mathbf{X}_{[-l^{(k+1)}:]} W^V W - \frac{1}{l^{(k)}} \mathbf{x}_0 W^Q (W^K)^\top \mathbf{x}_{[-l^{(k)}:]}^\top \mathbf{X}_{[-l^{(k)}:]} W^V W \right\|_2^2 \\ &= \left\| \mathbf{x}_0 W^Q (W^K)^\top \left( \frac{\mathbf{x}_{[-l^{(k+1)}:]}^\top \mathbf{X}_{[-l^{(k+1)}:]}}{l^{(k+1)}} - \frac{\mathbf{x}_{[-l^{(k)}:]}^\top \mathbf{X}_{[-l^{(k)}:]}}{l^{(k)}} \right) W^V W \right\|_2^2 \\ &\leq \left\| \mathbf{x}_0 W^Q (W^K)^\top \mathbf{x}_{[-(l^{(k)}-l):]}^\top \mathbf{X}_{[-(l^{(k)}-l):]} \left( \frac{1}{l^{(k+1)}} - \frac{1}{l^{(k)}} \right) W^V W \right\|_2^2 \\ &\quad + \left\| \mathbf{x}_0 W^Q (W^K)^\top \left( \frac{\mathbf{x}_{[-l^{(k+1)}:(l^{(k)}-l)]}^\top \mathbf{X}_{[-l^{(k+1)}:(l^{(k)}-l)]}}{l^{(k+1)}} - \frac{\mathbf{x}_{[-l^{(k)}:(l^{(k)}-l)]}^\top \mathbf{X}_{[-l^{(k)}:(l^{(k)}-l)]}}{l^{(k)}} \right) W^V W \right\|_2^2 \\ &\leq \frac{l_{\text{extra}}^2 (l^{(k)}-l)^2 C_0^2}{(l^{(k+1)} l^{(k)})^2} \\ &\quad + \left\| \frac{l + l_{\text{extra}}}{l^{(k+1)}} g_\theta^{l_{\text{train}}}(\mathbf{x}_{[-l^{(k+1)}:(l^{(k)}-l)]}) - \frac{l}{l^{(k)}} g_\theta^{l_{\text{train}}}(\mathbf{x}_{[-l^{(k)}:(l^{(k)}-l)]}) \right\|_2^2, \end{aligned} \tag{33}$$

where $C_0 = \left\| W^Q (W^K)^\top W^V W \right\|_2^2$ is a constant. The first term in (33) can be upper bounded by:

$$\frac{l_{\text{extra}}^2 (l^{(k)}-l)^2 C_0^2}{(l^{(k+1)} l^{(k)})^2} \leq \frac{l_{\text{test}}^2 C_0^2}{4 l_{\text{train}}^2}, \tag{34}$$

and for the second term in (33), we have:

$$\begin{aligned} & \left\| \frac{l + l_{\text{extra}}}{l^{(k+1)}} g_\theta^{l_{\text{train}}}(\mathbf{x}_{[-l^{(k+1)}:(l^{(k)}-l)]}) - \frac{l}{l^{(k)}} g_\theta^{l_{\text{train}}}(\mathbf{x}_{[-l^{(k)}:(l^{(k)}-l)]}) \right\|_2^2 \\ &\leq \left\| \frac{(l^{(k)}-l) l_{\text{extra}}}{l^{(k+1)} l^{(k)}} g_\theta^{l_{\text{train}}}(\mathbf{x}_{[-l^{(k+1)}:(l^{(k)}-l)]}) \right\|_2^2 \\ &\quad + \left( \frac{l}{l^{(k)}} \right)^2 \left\| g_\theta^{l_{\text{train}}}(\mathbf{x}_{[-l^{(k+1)}:(l^{(k)}-l)]}) - g_\theta^{l_{\text{train}}}(\mathbf{x}_{[-l^{(k)}:(l^{(k)}-l)]}) \right\|_2^2 \\ &\leq \frac{l_{\text{test}}^2}{4 l_{\text{train}}^2} + \left\| g_\theta^{l_{\text{train}}}(\mathbf{x}_{[-l^{(k+1)}:(l^{(k)}-l)]}) - g_\theta^{l_{\text{train}}}(\mathbf{x}_{[-l^{(k)}:(l^{(k)}-l)]}) \right\|_2^2. \end{aligned} \tag{35}$$

In fact, the second term above represents the alignment error between $\mathbf{x}_{[-l^{(k+1)}:(l^{(k)}-l)]}$ and $\mathbf{x}_{[-l^{(k)}:(l^{(k)}-l)]}$, where these two sequences have a length difference of $l_{\text{extra}}$.

On the other hand, for the third term in (32), if $l < l_{\text{train}}/2$, we have:

$$\left\|g_\theta^{l_{\text{train}}}(\mathbf{x}_{[-l_{\text{train}}:]}) - y(\mathbf{x}_{[-l_{\text{test}}:]})\right\|_2^2 \leq \left\|g_\theta^{l_{\text{train}}}(\mathbf{x}_{[-l_{\text{train}}:]}) - g_\theta^{l_{\text{train}}}(\mathbf{x}_{[-(l+l_{\text{train}}/2):]})\right\|_2^2$$
$$+ \left\|g_\theta^{l_{\text{train}}}(\mathbf{x}_{[-(l+l_{\text{train}}/2):]}) - g_\theta^{l_{\text{train}}}(\mathbf{x}_{[-l:]})\right\|_2^2 \qquad (36)$$
$$+ \left\|g_\theta^{l_{\text{train}}}(\mathbf{x}_{[-l:]}) - y(\mathbf{x}_{[-l_{\text{test}}:]})\right\|_2^2.$$

The first term above similarly represents the alignment error between two sequences with a length difference of $l_{\text{extra}}$, while the last term corresponds to the training error. Additionally, it is easy to derive an upper bound for the second term: $\left\|g_\theta^{l_{\text{train}}}(\mathbf{x}_{[-(l+l_{\text{train}}/2):]}) - g_\theta^{l_{\text{train}}}(\mathbf{x}_{[-l:]})\right\|_2^2 \leq 2$, and this upper bound also holds for the first term in (32). Meanwhile, if $l \geq l_{\text{train}}$, we have:

$$\left\|g_\theta^{l_{\text{train}}}(\mathbf{x}_{[-l_{\text{train}}:]}) - y(\mathbf{x}_{[-l_{\text{test}}:]})\right\|_2^2 \leq \left\|g_\theta^{l_{\text{train}}}(\mathbf{x}_{[-l_{\text{train}}:]}) - g_\theta^{l_{\text{train}}}(\mathbf{x}_{[-l:]})\right\|_2^2$$
$$+ \left\|g_\theta^{l_{\text{train}}}(\mathbf{x}_{[-l:]}) - y(\mathbf{x}_{[-l_{\text{test}}:]})\right\|_2^2. \qquad (37)$$

This result is similar to (36) except for lacking the second term in (36). Overall, we have:

$$\left\|g_\theta^{l_{\text{train}}}(\mathbf{x}_{[-l_{\text{test}}:]}) - y(\mathbf{x}_{[-l_{\text{test}}:]})\right\|_2^2 \leq (N_l + 2)\left(\sum_{k=0}^{N_l-1}\left\|g_\theta^{l_{\text{train}}}(\mathbf{x}_{[-l^{(k+1)}:(l^{(k)}-l)]}) - g_\theta^{l_{\text{train}}}(\mathbf{x}_{[-l^{(k)}:(l^{(k)}-l)]})\right\|_2^2\right.$$
$$+ \left\|g_\theta^{l_{\text{train}}}(\mathbf{x}_{[-l_{\text{train}}:]}) - g_\theta^{l_{\text{train}}}(\mathbf{x}_{[-(l+l_{\text{train}}/2):]})\right\|_2^2 \qquad \left(l < \frac{l_{\text{train}}}{2}\right)$$
$$+ \left\|g_\theta^{l_{\text{train}}}(\mathbf{x}_{[-l:]}) - y(\mathbf{x}_{[-l_{\text{test}}:]})\right\|_2^2$$
$$\left.+ N_l \frac{l_{\text{test}}^2(C_0^2 + 1)}{4l_{\text{train}}^2} + 4\right)$$

$$(38)$$

or

$$\left\|g_\theta^{l_{\text{train}}}(\mathbf{x}_{[-l_{\text{test}}:]}) - y(\mathbf{x}_{[-l_{\text{test}}:]})\right\|_2^2 \leq (N_l + 2)\left(\sum_{k=0}^{N_l-1}\left\|g_\theta^{l_{\text{train}}}(\mathbf{x}_{[-l^{(k+1)}:(l^{(k)}-l)]}) - g_\theta^{l_{\text{train}}}(\mathbf{x}_{[-l^{(k)}:(l^{(k)}-l)]})\right\|_2^2\right.$$
$$+ \left\|g_\theta^{l_{\text{train}}}(\mathbf{x}_{[-l:]}) - y(\mathbf{x}_{[-l_{\text{test}}:]})\right\|_2^2 \qquad \left(l \geq \frac{l_{\text{train}}}{2}\right).$$
$$\left.+ N_l \frac{l_{\text{test}}^2(C_0^2 + 1)}{4l_{\text{train}}^2} + 2\right)$$

$$(39)$$

It's easy to know that the probability of $l < l_{\text{train}}/2$ and $l \geq l_{\text{train}}/2$ is $\lfloor(l_{\text{train}} - 1)/2\rfloor/l_{\text{train}}$ and $\lfloor(l_{\text{train}} + 2)/2\rfloor/l_{\text{train}}$ respectively. Therefore, by taking the expectation over $l$ and $\mathbf{x}$ on both sides of the inequality, we have:

$$\mathcal{E}_{\text{gen}}(g_\theta^{l_{\text{train}}}; l_{\text{test}}) \leq \left(C_{l_{\text{test}}}^{(2)} + \frac{1}{l_{\text{train}}} \cdot \left\lfloor\frac{l_{\text{train}} - 1}{2}\right\rfloor\right)\mathcal{L}_{\text{misalign}}(g_\theta^{l_{\text{train}}}) + C_{l_{\text{test}}}^{(1)} \cdot \mathcal{L}_{\text{train}}(g_\theta^{l_{\text{train}}})$$
$$+ C_{l_{\text{test}}}^{(2)} \cdot \frac{l_{\text{test}}^2(C_0^2 + 1)}{4l_{\text{train}}^2} + C_{l_{\text{test}}}^{(1)} \cdot \frac{6l_{\text{train}} - 3 - (-1)^{l_{\text{train}}}}{2l_{\text{train}}}, \qquad (40)$$

where

$$C_{l_{\text{test}}}^{(1)} = \mathbb{E}_l[N_l + 2], C_{l_{\text{test}}}^{(2)} = \mathbb{E}_l[N_l(N_l + 2)]. \qquad (41)$$

It is easy to verify that $C_{l_{\text{test}}}^{(i)}$ increases as $l_{\text{test}}$ increases. Consequently, the constants $C_1^{(l_{\text{test}})} = C_{l_{\text{test}}}^{(2)} + \lfloor(l_{\text{train}} - 1)/2\rfloor/l_{\text{train}}, C_2^{(l_{\text{test}})} = C_{l_{\text{test}}}^{(1)}$ and $C_0^{(l_{\text{test}})} = C_{l_{\text{test}}}^{(2)}l_{\text{test}}^2(C_0^2 + 1)/(4l_{\text{train}}^2) + C_{l_{\text{test}}}^{(1)}(6l_{\text{train}} - 3 - (-1)^{l_{\text{train}}})/(2l_{\text{train}})$ also increases as $l_{\text{test}}$ increases. Moreover, the ratio $C_1^{(l_{\text{test}})}/C_2^{(l_{\text{test}})}$ increases as $l_{\text{test}}$ increases since the ratio $C_{l_{\text{test}}}^{(2)}/C_{l_{\text{test}}}^{(1)}$ increases as $l_{\text{test}}$ increases, which is easy to verify. $\square$

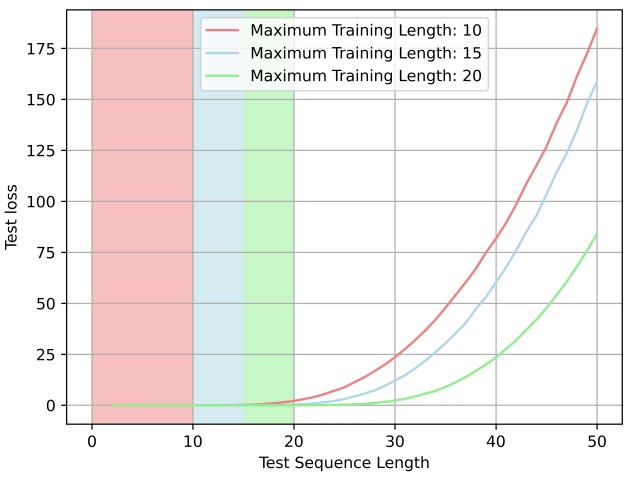
(a) Length generalization in sum prediction task with different maximum training sequence lengths

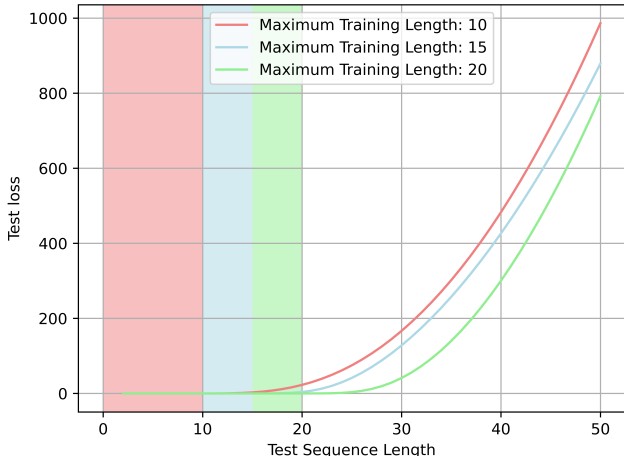
(b) Length generalization in length prediction task with different maximum training sequence lengths

Figure 4: Length generalization performance in the sum prediction and length prediction task with different maximum training sequence lengths. Although increasing the training length helps reduce the generalization error, the overall trend of increasing test loss remains unchanged.

## D. Analysis on the Sum Prediction Task

We also examine the sum prediction task, where the label corresponds to the sum of the sequence. Similar to the length prediction task, the output space shifts as the sequence length increases. As a result, models struggle with length generalization in this task, as shown in Figure 5a. However, by applying the reparameterization technique proposed in Section 3, we observe a significant improvement in length generalization, as shown in Figure 5b. These results demonstrate the importance of long-short alignment in length generalization.

## E. Pytorch-like Code for Implementation of $\mathcal{L}^*_{\text{train}}$

```python
# An efficient implementation for the total training loss.
import torch
import random

def SCE(output1_prob, output2_prob):
    loss = torch.mean((torch.sum(- output1_prob * torch.exp(output2_prob) - output1_prob *
        torch.exp(output1_prob), -1)))

extra_len = random.randint(1, max_len//2)
data = get_data(seq_len=max_len+extra_len)

output1 = model(data[:, :max_len])
output2 = model(data[:, -max_len:])
prob1 = torch.nn.functional.log_softmax(output1.logits)
prob2 = torch.nn.functional.log_softmax(output2.logits)

# Select the overlapped part to calculate the misalign loss
prob1 = prob1[:, max_len//2+extra_len:]
prob2 = prob2[:, max_len//2:max_len-extra_len]

loss_ce = (output1.loss + output2.loss) / 2
loss_misalign = SCE(prob1, prob2)

loss_total = loss_ce + alpha * loss_misalign
```

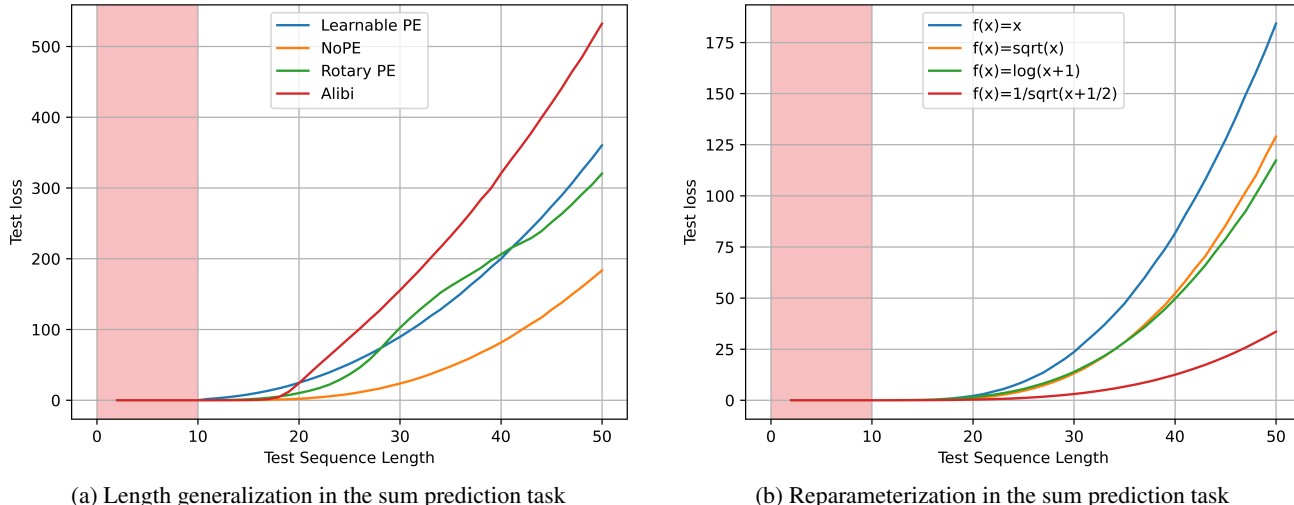

(a) Length generalization in the sum prediction task

(b) Reparameterization in the sum prediction task

Figure 5: Length generalization in the sum prediction task. Explicit alignment of output space boosts length generalization performance.

Table 9: Performance of the fine-tuned models using only cross-entropy loss (baseline) and an additional long-short misalignment loss on long-context modeling benchmark, LongBench-E score (Bai et al., 2023b) and perplexity on the 8k-length validation set. The fine-tuning sequence length is 4k, exactly the same as the training sequence length. We adopt two datasets: RedPajama-Book (Computer, 2023) and PG19 (Rae et al., 2019). The models finetuned with our proposed loss outperform the baseline across different model adaption strategies. The comparison is based on fixed total computation time. $(a/b)$ in the training steps mean $a$ steps for the baseline and $b$ steps for our method.

| Benchmark | LongBench-E ($\uparrow$) | | | Perplexity ($\downarrow$) | | |
| Training steps | 50/47 | 100/95 | 200/190 | 50/47 | 100/95 | 200/190 |
| --- | --- | --- | --- | --- | --- | --- |
| *RedPajama-Book* | | | | | | |
| $\mathcal{L}_{\text{train}}$ (Baseline) | 22.7 | 23.8 | 24.7 | 7.21 | 6.56 | 6.12 |
| $\mathcal{L}_{\text{train}} + 0.1\mathcal{L}_{\text{misalign}}$ (Ours) | **23.1** | **25.1** | **26.4** | **6.92** | **6.27** | **5.91** |
| $\mathcal{L}_{\text{train}} + 0.5\mathcal{L}_{\text{misalign}}$ (Ours) | 21.7 | 23.4 | 24.5 | 7.62 | 7.16 | 6.61 |
| *PG19* | | | | | | |
| $\mathcal{L}_{\text{train}}$ (Baseline) | 20.2 | 21.4 | 22.5 | **8.92** | **7.89** | 7.45 |
| $\mathcal{L}_{\text{train}} + 0.1\mathcal{L}_{\text{misalign}}$ (Ours) | **20.7** | **22.0** | **25.1** | 9.02 | 8.01 | **7.39** |
| $\mathcal{L}_{\text{train}} + 0.5\mathcal{L}_{\text{misalign}}$ (Ours) | 19.6 | **22.0** | 23.3 | 9.82 | 8.62 | 8.29 |

## F. Comparison Under Same Computation Time

To account for the additional computation time required to calculate $\mathcal{L}_{\text{misalign}}$ we will compare our methods with the baseline as in Table 2 and Table 3 based on total computation time. we compare our method with the baseline under equivalent total computation costs. Our method introduces an additional computational overhead of approximately $3\%$ to $5\%$ per step. To ensure fairness, we adjust the number of training steps proportionally. For example, when the baseline is trained for 50, 100, and 200 steps, our method is trained for 47, 95, and 190 steps, respectively, achieving comparable total computation times. We observe that the performance trends remain consistent: our method continues to outperform the baseline under equivalent computation time. This underscores the efficiency of our approach despite the minor additional cost.

## G. Additional Experiment on Mean Prediction Task with a Different Dataset Setting

In the synthetic experiments in Section 3, 0 and 1 have an equal probability. The mean value of 50 such samples will nearly obey the normal distribution $\mathcal{N}(0.5, 0.005)$. This means predicting 0.5 under a length of 50 would yield an approximate test loss of 0.005. However, we would like to clarify that the test loss of NoPE remains around 1e-5 (indicated by the orange line in Figure 1(a)), which is two orders of magnitude smaller than 0.005. This indicates that the model predicts the mean values

Table 10: Performance of the finetuned models using only cross-entropy loss (baseline) and an additional long-short misalignment loss on long-context modeling benchmark, LongBench-E score (Bai et al., 2023b) and perplexity on the 8k-length validation set. The fine-tuning sequence length is 8k. We adopt two kinds of model adjustments: LongQLora (Yang, 2023) and EABF (Zhang et al., 2024b). The models finetuned with our proposed loss outperform the baseline across different model adaption strategies. The comparison is based on fixed total computation time. $(a/b)$ in the training steps mean $a$ steps for the baseline and $b$ steps for our method.

| Benchmark | LongBench-E ($\uparrow$) | | | Perplexity ($\downarrow$) | | |
|---|---|---|---|---|---|---|
| Training steps | 50/47 | 100/95 | 200/190 | 50/47 | 100/95 | 200/190 |
| *LongQLora* | | | | | | |
| $\mathcal{L}_{\text{train}}$ (Baseline) | **21.9** | 22.1 | 23.4 | 6.82 | **6.41** | 5.82 |
| $\mathcal{L}_{\text{train}} + 0.1\mathcal{L}_{\text{misalign}}$ (Ours) | 21.6 | 23.1 | **25.7** | **6.79** | 6.42 | **5.74** |
| $\mathcal{L}_{\text{train}} + 0.5\mathcal{L}_{\text{misalign}}$ (Ours) | 21.1 | **23.6** | 24.9 | 7.19 | 6.65 | 5.96 |
| *EABF* | | | | | | |
| $\mathcal{L}_{\text{train}}$ (Baseline) | 22.1 | 22.9 | 23.6 | **6.89** | 6.52 | 6.01 |
| $\mathcal{L}_{\text{train}} + 0.1\mathcal{L}_{\text{misalign}}$ (Ours) | **23.0** | **23.6** | **24.5** | 7.01 | **6.48** | **5.86** |
| $\mathcal{L}_{\text{train}} + 0.5\mathcal{L}_{\text{misalign}}$ (Ours) | 22.2 | 23.1 | 23.8 | 7.32 | 6.88 | 6.42 |

of the sequences with high precision, rather than simply guessing a fixed number. Therefore, the conclusion that the model can have good length generalization ability in the mean prediction task is reasonable.

Besides, in order to avoid a trivial solution of predicting 0.5, we conduct an additional experiment on the mean prediction task. To build the sample of length $l$, we first sample the number of 1 uniformly from $[0, l]$ and then randomly build the sequence. In this way, the mean value of the sequence uniformly spans from 0 to 1, avoiding trivial prediction. The experimental results are shown in Figure 6, where the model can still achieve good length generalization, consistent with our previous results.

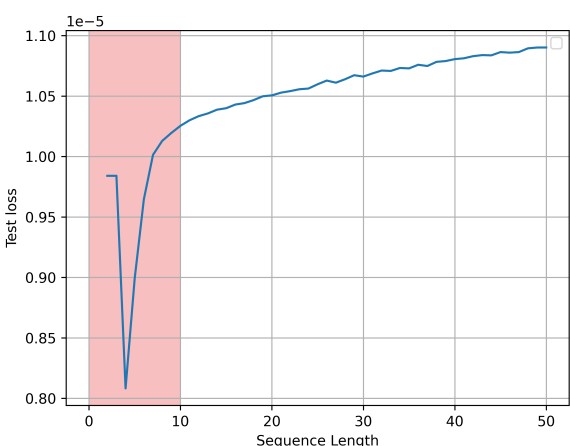

Figure 6: Length generalization in the mean prediction task with a different dataset setting. To build the sample of length $l$, we first sample the number of 1 uniformly from $[0, l]$ and then randomly build the sequence. In this way, the mean value of the sequence uniformly spans from 0 to 1, avoiding trivial prediction. The model can still achieve good length generalization, which is consistent with our previous results.

