# OpenReview forum: "Long-Short Alignment for Effective Long-Context Modeling in LLMs"
_ICML.cc/2025/Conference — ICML 2025 poster_

### Official Review · Reviewer_qzDj · 2025-02-17

**Overall Recommendation:** 3

**Summary:**

This work suggests matching the long-sequence loss and the short-sequence loss, named output alignment. The author has conducted experiments to prove the effectiveness of the proposed methods.

**Claims And Evidence:**

The work supports the claims and evidence.

**Essential References Not Discussed:**

N/A

**Experimental Designs Or Analyses:**

I have checked the soundness/validity of experimental designs and analyses

**Methods And Evaluation Criteria:**

The proposed methods evaluation criteria make sense. However, there are still some questions:
* **Figure 1 (a), the NoPE loss is almost zero**. Is anything wrong?
* **The proposed method may degrade the performance for the shorter length, which is proved in Table 2 and Table 3.**
* **The method is sensitive to hyperparameters for the loss misalign**, which is proved in Table 6.

**Other Comments Or Suggestions:**

M/A

**Other Strengths And Weaknesses:**

N/A

**Questions For Authors:**

The proposed method is sensitive to the hyperparameter for loss misalign (proved in Table 6). Is there any way to help choose the hyperparameter?

**Relation To Broader Scientific Literature:**

This works propose

**Theoretical Claims:**

I have checked the proofs.

---

> ### Author Rebuttal · Authors · 2025-04-01
>
> We thank Reviewer qzDj for the comments. We will address your questions in the following part.
>
> ---
> Q1. Figure 1 (a), the NoPE loss is almost zero. Is anything wrong?
>
> A1. Thank you for your question. The NoPE loss is indeed almost zero (around 1e-5). This observation is also supported by our theoretical results in Appendix C.1.
>
> ---
>
> Q2. The proposed method may degrade the performance for the shorter length, which is proved in Table 2 and Table 3.
>
> A2. Thank you for your feedback. We apologize for the potential misunderstanding in Table 2 and 3. Table 2 and Table 3 do not evaluate performance with respect to sequence length. Instead, they show that our proposed method may experience a slight degradation compared to the baseline when trained for only 50 epochs (8.92 v.s. 8.95 in Table 2, 6.89 v.s. 6.92 in Table 3). Importantly, as shown in the same tables, our proposed method benefits significantly from longer training (100 and 200 epochs), demonstrating the effectiveness with sufficient training.
>
> ---
>
> Q3. The method is sensitive to hyperparameters for the loss misalign, which is proved in Table 6. Is there any way to help choose the hyperparameter?
>
> A3. Indeed, the misalignment loss is a regularization term, and like many regularization techniques, it can influence the training process—a phenomenon observed in prior work [1, 2]. As mentioned in Section 5.4 (Line 412), we suggest using a coefficient $\alpha$ between 0.1 and 0.3 as a default to mitigate the risk of over-regularization while maintaining performance.
>
> [1] Zhao et al, When Will Gradient Regularization Be Harmful? ICML 2024.
>
> [2] Srivastava et al. Dropout: a simple way to prevent neural networks from overfitting. Journal of Machine Learning Research 2014.
>
> ---
>
> Thanks again for your comments, and hope our response could address your concerns. Please let us know if you have additional questions.

---

> > ### Comment · Reviewer_qzDj · 2025-04-05
> >
> > Thank you very much for the response. I would like to further discuss the Q2: Table 2 and Table 3.
> >
> > **Length Extrapolation Problem**
> > Currently, the length extrapolation is difficult because the Transformer cannot handle long-sequence. For example, if a model is trained for a length of 1024, and then we use it to process on length of 4096.
> > * Choice 1: abandon the first 3072 tokens and only use the last 1024 tokens to predict the next token.
> > * Choice 2: Use the whole 4096 tokens to predict the next token.
> > * If we use RoPE and validate it on language modeling, we will find that the Choice 1 PPL is lower than the Choice 2 PPL [1-2].
> > * **For this work:** it actually forces to alignment of choice 1 and choice 2.
> >
> > **To improve the score: re-evaluate baseline**
> > * **Evaluate baseline within training length 4096**. For example, evaluate with lengths 1024, 2048, and 4096. We conduct this experiment to check whether the proposed method degrades the performance within the training length.
> > * **The baseline and proposed method performance without CLEX**. The CLEX has a maximum extrapolation length, which is actually similar to randomized position encoding that let the model sees all the potential position IDs within the maximum extrapolation length. However, the model still cannot achieve a good length of extrapolation performance beyond the maximum extrapolation length.
> >
> >
> > Reference:
> >
> > [1] Fang, L., Wang, Y., Liu, Z., Zhang, C., Jegelka, S., Gao, J., ... & Wang, Y. (2024). What is Wrong with Perplexity for Long-context Language Modeling?. arXiv preprint arXiv:2410.23771.
> >
> > [2] Zheng, C., Gao, Y., Shi, H., Xiong, J., Sun, J., Li, J., ... & Li, Y. (2024). DAPE V2: Process Attention Score as Feature Map for Length Extrapolation. arXiv preprint arXiv:2410.04798.

---

> > > ### Author Response · Authors · 2025-04-09
> > >
> > > **Extra Q1. (Length Extrapolation Problem)** About the two choices.
> > >
> > > **A1.** Thank you very much for your insightful comment on the comparison. We understand your point to be that Choice 1 often yields lower perplexity than Choice 2, possibly because the model avoids extrapolating to unseen positional embeddings beyond the training length. In other words, while Choice 2 can use more information, it also introduces a **distribution shift in positional representations**, which may lead to degraded performance. In fact, this behavioral gap between Choice 1 and Choice 2 highlights a fundamental challenge in length extrapolation: **model predictions can become inconsistent depending on the portion of the context that is used**.
> > >
> > > Our method addresses exactly this issue: by aligning the model’s output distributions across inputs of different lengths, we encourage **prediction consistency between Choice 1 and Choice 2**. To empirically validate this, we evaluated the model under all three settings on LongBench-E:
> > >
> > > - **Setting 1 (Choice 1)**: Truncate the query to 4096 tokens. Use the baseline model.
> > > - **Setting 2 (Choice 2)**: Use the full query. Use the baseline model.
> > > - **Setting 3 (Ours)**: Use the full query. Use our proposed method.
> > >
> > > The results are shown in the following table:
> > >
> > > |  | Setting 1 | Setting 2 | Setting 3 |
> > > | --- | --- | --- | --- |
> > > | LongBench-E Score | 18.2 | 8.9 | 26.6 |
> > >
> > > The key result is that our method significantly outperforms both Setting 1 and 2. This shows that our approach not only enables the model to effectively utilize the full long context, but also surpasses the best workaround (truncation) in performance.
> > >
> > > ---
> > >
> > > **Extra Q2. (Evaluate baseline within training length 4096)** Evaluate with lengths 1024, 2048, and 4096. We conduct this experiment to check whether the proposed method degrades the performance within the training length.
> > >
> > > **A2.** Thank you for your helpful suggestion. We conduct additional experiments following the setting of Table 2, using a training length of 4096 and evaluating perplexity at context lengths 1024, 2048, and 4096 after 200 steps of training. The results are shown below:
> > >
> > > |  | Length=1024 | Length=2048 | Length=4096 |
> > > | --- | --- | --- | --- |
> > > | Baseline | 6.67 | 6.12 | 5.74 |
> > > | Ours | 6.62 | 6.08 | 5.81 |
> > >
> > > The results indicate that our proposed method does not degrade performance at a shorter length. Intuitively, our proposed regularization does not penalize or distort the model's ability to model short sequences, which behaves as a gentle consistency constraint rather than an extrapolative bias.
> > >
> > > ---
> > >
> > > **Extra Q3. (The baseline and proposed method performance without CLEX**.**)** The CLEX has a maximum extrapolation length.
> > >
> > > A3. Thank you for your insightful comment. We fully agree that CLEX, like many other extrapolation methods, has a theoretical upper bound on extrapolation length. This limitation arises from the design principle behind CLEX and similar approaches—namely, exposing the model to a range of position IDs during training so that it can generalize within that range. This principle is also adopted in other state-of-the-art extrapolation methods, including PI [1], ABF [2], NTK-Aware [3] (used in CodeLlama [4]), Yarn [5], EABF [6], LongQLora [7], and CREAM [8].
> > >
> > > As you suggested, we also evaluated our method in settings **without CLEX**, and the results are included in Table 3 of our paper (EABF, LongQLora) and A1 to Reviewer Ueg1 of our rebuttal (CREAM). For example, the table below shows results from combining our method with CREAM:
> > >
> > > |  | LongBench-E | Perplexity |
> > > | --- | --- | --- |
> > > | CREAM | 23.6 | 6.62 |
> > > | Our method + CREAM | 25.2 | 5.94 |
> > >
> > > These results suggest that although existing methods have extrapolation limits, our approach can complement them by improving alignment across lengths, which is **orthogonal** to the positional encoding design itself.
> > >
> > > [1] Chen et al. Extending context window of large language models via positional interpolation. arXiv:2306.15595.
> > >
> > > [2] Xiong et al. Effective long-context scaling of foundation models. arXiv:2309.16039.
> > >
> > > [3] URL https://www.reddit.com/r/LocalLLaMA/comments/14lz7j5/ntkaware_scaled_rope_allows_llama_models_to_have/
> > >
> > > [4] Rozière et al. Code Llama: Open Foundation Models for Code. arXiv:2308.12950
> > >
> > > [5] Peng et al. Yarn: Efficient context window extension of large language models. arXiv:2309.00071.
> > >
> > > [6] Zhang et al. Extending llms’ context window with 100 samples. arXiv:2401.07004.
> > >
> > > [7] Yang et al. Longqlora: Efficient and effective method to extend context length of large language models. arXiv:2311.04879
> > >
> > > [8] Wu et al. An Efficient Recipe for Long Context Extension via Middle-Focused Positional Encoding. NeurIPS 2024.
> > >
> > > ---
> > >
> > > Thanks again for your time and thoughtful feedback! We hope our response addresses your concerns. If you find our clarifications and additional results satisfactory, we would be grateful if you would consider updating your evaluation accordingly.

---

### Official Review · Reviewer_F5vG · 2025-02-18

**Overall Recommendation:** 3

**Summary:**

This paper targets **length generalization** problem for LLMs and proposes to **shift from** conventional perspective of **positional encodings and data structures to the output distribution** of the model. They argue that the consistency of output distributions across sequences with different length correlates well with length generalization performance. They name this consistency **outer alignment** and propose a metric called **Long-Short Misalignment** to quantify it, and further design a **regularization loss** to explicitly enhance outer-alignment. They support their claim both empirically and theoretically.

**Claims And Evidence:**

- **Concern 1**: my biggest concern is on the difference between mean prediction and length prediction task. The author points out in L160-164 that (1) for mean prediction task, the output remain in $[0,1]$ and (2) for length prediction task, the output support set grows with longer length, (3) this difference motivates them to consider output alignment. However, **there exists many tasks whose output support set do not grow with longer length, see Delétang et al 2022.** Transformers still struggle on these tasks. This makes me question if the motivation is valid.

- **Concern 2**: On the connection between synthetic task and language modeling task. Using the mean prediction and length prediction task (Section) to motivate Section 4 seems not good, since the former is essentially a **regression task** (predicting continuous target) while the author extend their claim to **sequence modeling** subsequently.

Reference:
Delétang et al 2022. Neural Networks and the Chomsky Hierarchy.

**Essential References Not Discussed:**

I believe the authors have discussed all necessary references.

**Experimental Designs Or Analyses:**

- **Experimental Design 1**: In Section 4.1, the $l_1$ and $l_2$ are sampled from $[l_{train}/2, l_{train}]$ and $l_{train}$ is the model's training context length. This can be a large range, which defeats the claim (L064, L216) that the $x_{[-l_1:]}$ and $x_{[-l_2:]}$ are only **slightly** different? **The motivation of output distribution being similar should only hold when the length differences are indeed small.** Also, in L431, the author mentioned that aligning output distributions with **moderate** length discrepancies improves. I think the author should be more careful on the consistency of their claims.

- **Minor**:
    - In L593 (Appendix B), I was confused on how the hyper-parameters is selected, could you clarify?
    - In Section 3, why are those transformation selected?
    - Again in Section 3, what loss are used to train the model? My guess is MSE (seen from Equation 8) but this should be more explicitly mentioned.

**Methods And Evaluation Criteria:**

- **Evaluation criteria 1**: The author proposes a new perspective of improving length generalization but **does not show clearly how important this perspective is**. While it makes sense to compare the proposed regularization with naive fine-tuning, the paper would benefit a lot by comparing it with some representative length generalization methods. Namely, this can answer the question that should the community considers more on this direction because it improves more than conventional perspectives (e.g. positional encodings)? However, I admit that this might be out of scope and is just out of my personal curiosity.

**Other Comments Or Suggestions:**

Minor comments:

- **Comment 1**: Presentation can be a bit smoother. For example, the first paragraph of Section 3 can be much shorter since the related work and background was just discussed above.
- **Comment 2**: The title of Table 7 seems wrong: it should be ablation study on 'sampling range'.

**Other Strengths And Weaknesses:**

Summary of strengths and weaknesses.
- **Strength**: the idea to study length generalization on output distributions is novel, and the key claims are supported both theoretically and empirically.
- **Weakness**: the motivation obtained from the synthetic tasks does not make a lot of sense to me, and the connection between synthetic task and language modeling task, as well as their corresponding proposed approaches (i.e. Section 3 and 4) is weak, in the paper's current format.

**Questions For Authors:**

Most of my questions have been mentioned above. I would be happy to increase my score if the authors can address/clarify **Concern 1&2, Experimental design 1** and **Evaluation criteria 1**. For example, I suggest the following, respectively (the author can provide more suitable clarifications if there are any)
- **Concern 1**: I don't have a concrete idea on how Concern 1 can be addressed. I let the authors decide.
- **Concern 2**: discuss more on how explicit reparametrization relates to the regularization loss, and how section 3 connects with section 4.
- **Experimental design 1**: Refine and rationalize their claim on which lengths should the consistency be enforced on, should the length differences be slight, moderate, or something else? Given $l_{train}$ can be large, maybe the author can provide experiments on something like $[0.95*l_{train}, l_{train}]$.
- **Evaluation criteria 1**: Add a preliminary experimental comparison between their approach and some representative length generalization approaches (e.g. compare to length-extrapolatable position encodings).

**Relation To Broader Scientific Literature:**

- **Broader literature 1**: The regularization loss to enforce output distribution consistency between lengths broadly connects to **better fine-tuning strategies**.
- **Broader literature 2**: The idea of keeping output distributions similar for similar inputs (here in terms of length) is distantly related to keeping model output similar on inputs with data augmentations (e.g. a rotated image), which is in general used in **Test-time training literature**.

**Theoretical Claims:**

I checked the Theorem 4.1 (which should be the only theoretical claim), and did not find a clear problem. However, it is possible that there is a miss.

---

> ### Author Rebuttal · Authors · 2025-04-01
>
> Q1. About Delétang et al. 2022. This makes me question if the motivation is valid.
>
> A1. Thank you for your insightful comment. We agree that the difficulty of achieving length generalization in Transformers likely stems from multiple factors. In the related work section, we have acknowledged that positional encoding plays a significant role in this challenge. However, our findings suggest that output misalignment is another important factor, as evidenced by our comparisons.
>
> To clarify, we do not claim that output misalignment is the sole reason for poor length generalization. Rather, we identify it as one contributing factor and show that addressing it leads to improved generalization. While some tasks may not exhibit an expanding output support set with increasing length, our focus is on cases where such a shift does occur, as it presents a clear source of distributional mismatch.
>
> Moreover, our results demonstrate that mitigating output misalignment leads to better length generalization, even if it does not fully resolve the problem. This suggests that output alignment is a meaningful component of the broader challenge, rather than an all-encompassing solution. We will clarify this distinction in the final version.
>
> ---
>
> Q2. On the connection between synthetic task and language modeling task.
>
> A2. We acknowledge that the tasks in the two sections are not the same type. However, our motivation for including Section 3 is not to claim that the two tasks are identical, but rather to provide a controlled setting that isolates one specific challenge in length generalization: **output misalignment**.
>
> While sequence modeling is more complex, the fundamental issue remains—**the output distribution can shift as input length varies**, which can harm generalization. Section 3 demonstrates this effect in a simpler setting where the misalignment can be analyzed more clearly. This serves as motivation for Section 4, where we propose methods to mitigate similar issues in the more complex sequence modeling setting. We will refine the discussion to better bridge the gap between the synthetic tasks and sequence modeling.
>
> ---
>
> Q3. The paper would benefit a lot by comparing it with some representative length generalization methods.
>
> A3. Thank you for the suggestion. Since most representative length generalization methods modify RoPE, we compare our method applied on top of EABF against CREAM [1], another method that builds on EABF. The results for 100 steps training are shown in the table below.
>
> |  | LongBench-E | Perplexity |
> | --- | --- | --- |
> | Our method | 24.0 | 6.43 |
> | CREAM | 23.6 | 6.62 |
>
> We find that our method outperforms CREAM in both LongBench-E score and perplexity, suggesting that output alignment provides additional benefits beyond positional encoding modifications.
>
> [1] Wu et al. An Efficient Recipe for Long Context Extension via Middle-Focused Positional Encoding. NeurIPS 2024.
>
> ---
>
> Q4. Refine and rationalize their claim on which lengths should the consistency be enforced on. The author can provide experiments on something like $[0.95*l_{train},l_{train}]$.
>
> A4. Thank you for pointing this out. We agree that "moderate" would better reflect our intended meaning in L064 and L216. Our goal is to enforce consistency across **moderate** length differences rather than extremely small ones, as small variations may not provide sufficient regularization. We will refine the wording in the paper to better reflect this insight.
>
> Following your suggestion, we also conduct experiments using the sampling range $[0.95*l_{train},l_{train}]$ (which implies that the sampling range of $l_{extra}$ is $[1, 0.05l_{train}]$) following the setting in Table 7. The results of 200 steps are shown in the table below.
>
> | Benchmark | LongBench-E | Perplexity |
> | --- | --- | --- |
> | Baseline | 23.4 | 5.82 |
> | $[1, l_{train}/2]$ | 25.8 | 5.77 |
> | $[1, 0.05l_{train}]$ | 23.9 | 5.87 |
>
> While restricting $l_{extra}$ to $[1, 0.05 \cdot l_{train}]$ still provides some improvement, the broader range $[1, l_{train}/2]$ is more effective. This suggests that keeping consistency over moderate rather than minimal length differences is beneficial.
>
> ---
>
>
> Q5. I was confused on how the hyper-parameters is selected, could you clarify?
>
> A5. Sure. The batch size is 256. We use SGD as the optimizer and search for the initial learning rate in {5e-5, 1e-4, 5e-4}. The learning rate follows a cosine schedule. The models are trained for 40,000 epochs. The training loss is Mean Squared Error (MSE).
>
> ---
>
> Q6. In Section 3, why are those transformation selected?
>
> A6. To align the output distribution across different input lengths, $f$ is chosen based on the following criteria: (1) $f$ must be a bijection to ensure the existence of $f^{-1}$ (2) $f$ should be a contraction mapping on $[1, +\infty]$ to reduce the discrepancy in outputs across different input lengths.
>
> ---
>
> Also thanks for the minor comments! We will address them in the final version.

---

### Official Review · Reviewer_Ueg1 · 2025-03-12

**Overall Recommendation:** 4

**Summary:**

Authors introduce a novel perspective on length generalization in large language models (LLMs) by focusing on output alignment rather than conventional input-based approaches like positional encodings. Through synthetic task case studies, the authors demonstrate that models generalize better when output distributions remain consistent across varying sequence lengths. They propose the Long-Short Misalignment metric to quantify output alignment and introduce a regularization loss to enhance length generalization. Extensive experiments on synthetic and natural language tasks validate their approach, offering insights into improving LLMs' performance on longer contexts.

**Claims And Evidence:**

The claims in the paper are generally well-supported by empirical and theoretical evidence.

**Essential References Not Discussed:**

No

**Experimental Designs Or Analyses:**

Yes, the experimental setup is okay, but I would rather see more comparison with other methods.

**Methods And Evaluation Criteria:**

Yes, I would rather also see more comparison with other methods for improving length generalization.

**Other Comments Or Suggestions:**

See questions.

**Other Strengths And Weaknesses:**

Strength:

* focusing on output alignment for length generalization which I believe can be another axes to other methods like positional encodings.
* Providing synthesis task analysis to bake the intention for following the idea.
* Generally speaking, I think this is an insightful paper.


Weakness:
* I believe comparison with other methods in length generalization is needed. Also seeing if this technique can be combined with other techniques to see if it further improves the performance.

**Questions For Authors:**

Question1: In the length generalization experiment, why do you think 1 over square root function works the best? Is it just because the values are in [0,1] range? what about other functions that puts the output distribution in this range? Do they work as well?

Question 2: I believe more comparison with other length generalization approaches can help the results. As well if combining those methods with yours helps would be one of my questions.

Question 3: What positional encoding is used for main results like in table 1? Do you think if changing the positional encoding might affect the results? A comparison of them against each other might be helpful. Especially when the initial positional encoding is not good at length generalization and your approach improve length generalization which means your approach can be stand alone for length generalization.

**Relation To Broader Scientific Literature:**

The paper builds on prior work on length generalization in Transformers, which has primarily focused on positional encodings and modeling mechanisms. Unlike these studies, it shifts the focus to output alignment, introducing the Long-Short Misalignment metric as a new predictor of generalization and proposing a regularization loss to enhance performance.

**Theoretical Claims:**

Yes, and particularly I am more interested on empirical results as I believe they are more important in the context of this research.

---

> ### Author Rebuttal · Authors · 2025-04-01
>
> We thank Reviewer Ueg1 for appreciating the insight of our paper. We will address your questions in the following part.
>
> ---
> Q1. (From **Other Strengths And Weaknesses**) I believe comparison with other methods in length generalization is needed. Also seeing if this technique can be combined with other techniques to see if it further improves the performance.
>
> A1. Thank you for your suggestion. To clarify, both Table 2 and Table 3 compare our proposed method combined with a length generalization technique against the length generalization technique alone. The results demonstrate that our method consistently improves performance when combined with existing approaches.
>
> To further address your point, we compare our method applied on top of EABF against CREAM [1], another method built on EABF. Additionally, we evaluate whether combining our method with CREAM leads to further improvements. The results for 100-step training are shown below:
>
> |  | LongBench-E | Perplexity |
> | --- | --- | --- |
> | Our method | 24.0 | 6.43 |
> | CREAM | 23.6 | 6.62 |
> | Our method + CREAM | 25.2 | 5.94 |
>
> These results show that our method outperforms CREAM alone in both LongBench-E score and perplexity. Furthermore, combining our method with CREAM leads to additional improvements. This supports our claim that output alignment plays a crucial role in length generalization and can complement existing approaches.
>
> [1] Wu et al. An Efficient Recipe for Long Context Extension via Middle-Focused Positional Encoding. NeurIPS 2024.
>
> ---
>
> Q2. (From **Questions For Authors**) In the length generalization experiment, why do you think 1 over square root function works the best? Is it just because the values are in [0,1] range? What about other functions that puts the output distribution in this range? Do they work as well?
>
> A2. We conducted additional experiments using alternative parameterization functions, including $f(x)=1/\log(x+1)$ and $f(x)=1/x$ following your suggestions. The results, presented in the anonymous link https://ibb.co/39M3D8Nq, show that while these functions also improve performance, $f(x)=1/\sqrt{x}$ still performs the best.
>
> The superior performance of $f(x)=1/\sqrt{x}$ is not solely due to its range being within [0,1]. Rather, we hypothesize that it is because, both empirically (Figure 1(b)) and theoretically (Theorem C.1), the test loss is proportional to the square of the input length when no parameterization function $f$ is applied. Therefore, using $f(x)=1/\sqrt{x}$ effectively normalizes the output and mitigates this issue.
>
> ---
>
> Q3. (From **Questions For Authors**) What positional encoding is used for main results like in table 1? Do you think if changing the positional encoding might affect the results? A comparison of them against each other might be helpful. Especially when the initial positional encoding is not good at length generalization and your approach improve length generalization which means your approach can be stand alone for length generalization.
>
> A3. Thank you for your question. For the main results in Table 1, GPT-J-6B, GPT-NeoX-20B, Llama2-7B, and Qwen-7B-8K use the original RoPE, while Yarn-Llama2-7B-8K and CLEX-Llama-4K use modified RoPE and achieve better length generalization.
>
> Changing the positional encoding can indeed affect the results, as seen in Table 1—models using modified RoPE tend to perform better in terms of length generalization. However, even among models using the same RoPE (e.g., GPT-J-6B, GPT-NeoX-20B, Llama2-7B, and Qwen-7B-8K), we observe varying degrees of length generalization. This suggests that while positional encoding plays a role, it is not the sole factor.
>
> As shown in Table 11, our method does not lead to significant performance gains when using the original RoPE, likely because using original RoPE will lead to slow convergence speed in length generalization task [1, 2]. However, when we use modified RoPE, our method provides notable improvements in length generalization, as shown in Table 1. This suggests that while the choice of positional encoding does impact performance, our method works best when combined with a modified version of RoPE that better supports length generalization.
>
> [1] Chen et al. Extending Context Window of Large Language Models via Positional Interpolation. arxiv 2306.15595
>
> [2] Zhang et al. Extending LLMs' Context Window with 100 Samples. arxiv 2401.07004
>
> ---
>
> Thanks again for your comments, and hope our response could address your concerns. Please let us know if you have additional questions.

---

### Official Review · Reviewer_KAyJ · 2025-03-13

**Overall Recommendation:** 3

**Summary:**

This paper studies how aspects of the output distributions of LLMs relate to length generalization and performance on long-context tasks. First, the authors show that train-test mismatch in the output space can lead to poor generalization on synthetic tasks, and that reformulating the tasks to reduce this mismatch improves performance. Second, on non-synthetic natural language tasks, the authors propose a metric called "long-short misalignment" which measures the divergence between the output distributions of a model when it's original context window is truncated (up to 50%). This measures the model's invariance to this truncation. There is a positive correlation between models that are more invariant (and therefore less sensitive to earlier context) and performance on long-context modeling tasks. The authors then propose a regularizer based on this metric, which when applied during training can improve performance on tasks such as LongBench-E.

## Update after rebuttal

I would strongly suggest including the expanded Table 5 in the revised version, and reducing Section 3 to improve the presentations, as proposed in the response. While still a bit unintuitive to me (I would have thought more tasks require sensitivity to earlier context) the empirical results seem reasonably strong across most benchmarks, and the authors show that hyperparameters can be chosen such that sensitivity to earlier context is not entirely eliminated.

Assuming the pledged changes will be implemented, I will increase my score from 2 to 3.

**Claims And Evidence:**

The main empirical claims seem to be supported. The proposed regularizer improves performance on benchmarks such as LongBench-E.

**Essential References Not Discussed:**

This is not critical, just a connection that seemed relevant as I was reading the paper:

- There is a large body of work in NLP related to developing new methods and Transformer variants to improve length generalization e.g. on length-based splits of SCAN (https://arxiv.org/abs/1711.00350). A focus in this line of work has been on understanding the relation between the degree of context sensitivity and out-of-distribution generalization. For example, various approaches have injected CFG-like biases that reduce context sensitivity. The methods that the authors propose in the submitted paper seems related in that they attempt to reduce context sensitivity, in this case encouraging models to be invariant to context earlier in the context window. Maybe there is an interesting connection to be discussed.

**Experimental Designs Or Analyses:**

The experimental design seems reasonable.

**Methods And Evaluation Criteria:**

The proposed benchmarks seem reasonable.

**Other Comments Or Suggestions:**

* nit: It would be helpful to make the Figures easier to read without color, i.e. use different line shadings or styles in Figure 1.
* One of the models looks very underfit in Figure 1a, perhaps worth investigating the training configuration.
* The introduction frames the proposed method as a completely different perspective from prior work investigating alternative positional encodings. However, it seems both lines of work have considered ways to reduce the sensitivity of models to potentially irrelevant context, e.g. the biases of methods such as RoPE that discourage long-range dependencies. It could be interesting to see if different positional encoding schemes lead to different degrees of "long-short misalignment" in the predictable way.

**Other Strengths And Weaknesses:**

Strengths:

- The paper studies how a metric related to how invariant a model's outputs are to truncating the input context predicts performance on various long-context and length generalization settings.
- Inspired by this, the authors propose a regularizer that encourages model outputs to be invariant to truncating early context. This appears to improve generalization performance on several tasks.

Weaknesses:

My main concerns are related to the paper presentation, which I think could be considerably improved. I think it would also be useful to better understand the weaknesses of the method, since the proposed regularization (which encourages models to ignore early context) seems like it must surely be harmful in some settings. It would be good to clarify this so practitioners can have better intuition for when to apply the proposed method.

- Section 3 was a bit distracting to me as a reader. It is not surprising that reducing train and test mismatch in the output space is valuable for out-of-distribution generalization, including length generalization. The results did not seem very closely connected to the proposed methods in Section 4, despite the attempt of Table 8 to clarify this. Section 3 relates to a property of the *task*, i.e. re-formulating the task to reduce the train-test distribution shift of the outputs, and is well supported in theory. The methods in Section 4 relate to a property of the *model*, i.e. that it should be largely invariant to earlier context. This is less clearly useful from a theoretical perspective, and therefore a bit surprising that it is effective. I think the paper would be stronger without Section 3, at least in the main paper.
- The proposed terminology of "Output Alignment" is quite confusing. First, "Alignment" is overloaded with various techniques for model post-training, e.g. RLHF, and the proposed technique has nothing to do with this. The method is most clearly understood as encouraging invariance of output distributions when the context window is truncated, i.e. encouraging a form of context insensitivity.
- It would be good to understand the limitations of the proposed method. For example, in cases that require sensitivity to information early in the context window, the proposed regularization should intuitively harm performance. The ablations in section 5.3 could be expanded to highlight potential weaknesses of the method (see questions for authors).

**Questions For Authors:**

1. Section 5.3 - Intuitively, the proposed regularization should be harmful for cases where necessary context appears early in the context window, so it is surprising that the scores for "Fact Depth" 0% and 25% are comparable to the baseline. Does this change if the regularization coefficient is increased?

**Relation To Broader Scientific Literature:**

To the best of my knowledge, the proposed objective is novel.

**Theoretical Claims:**

There is a theoretical claim in 4.1, although it is only stated imprecisely in the main paper and I did not review the details in the appendix.

---

> ### Author Rebuttal · Authors · 2025-04-01
>
> Q1. Maybe there is an interesting connection to be discussed with SCAN.
>
> A1. Thanks for highlighting these works! Indeed, SCAN-based methods leverage CFG-like rules to generate training data that reduce context sensitivity, which can help improving out-of-distribution generalization. This is indeed relative to the concept behind our proposed output matching, which ensures semantic consistency across different input lengths. However, there are key differences:
>
> - Different Focus. SCAN-based approaches primarily operates from the input perspective by generating structured data, while our method focuses on output matching and introduces a fresh output perspective.
> - Better Efficiency. SCAN-based approaches need manual CFG design, making them time-consuming and less scalable, while our method can offer a more efficient solution with only ~5% additional computation.
>
> We will clarify this distinction and discuss potential connections to these prior works in the revised paper.
>
> ---
>
> Q2. The results in Section 3 did not seem very closely connected to the proposed methods in Section 4.
>
> A2. Thank you for your insightful comments. We clarify that Section 3 is crucial for our paper for the following reasons:
>
> - Fundamental Motivation for Length Generalization Strategies: Section 3 highlights that one of the key challenge in length generalization is that the output distribution shifts when input length changes. This sets the stage for why controlling output alignment is crucial.
> - Bridging Task Reformulation and Model-Based Regularization: Both Section 3 and Section 4 aim to reduce output distribution discrepancies across different input lengths. When we have a strong prior about the task, we can reformulate it directly (as in Section 3). However, when such a prior is unavailable, we instead introduce a regularization approach (Section 4) to encourage the model to implicitly learn this property. Without Section 3, the necessity of such regularization may be less clear. By including both, the paper offers a broader perspective on addressing length generalization, rather than focusing solely on a single heuristic.
>
> Thank you again for raising this point, we will make Section 3 more concise to avoid distracting the main contributions of Section 4.
>
> ---
>
> Q3. The proposed terminology of "Output Alignment" is overloaded with various techniques for model post-training.
>
> A3. Thanks for pointing it out! We will use “Output Matching” instead to avoid such confusion.
>
> ---
>
> Q4. It would be good to understand the limitations of the proposed method. The ablations in section 5.3 could be expanded to highlight potential weaknesses of the method. Does this change if the regularization coefficient is increased?
>
> A4. Thanks for your suggestions! We acknowledge the importance of discussing the limitations of our method and have addressed this in Section 5.4 (line 408). Following your suggestions, we conduct additional experiments to further analyze the effect of $\alpha$ based on the ablations in Table 5. Specifically, we extend the range of $\alpha$ to include 0, 0.1, 0.3 and 0.5. The results are shown in the table below.
>
> | $\alpha$ | Depth=0% | Depth=25% | Depth=50% | Depth=75% |
> | --- | --- | --- | --- | --- |
> | 0 | 75 | 64 | 30 | 69 |
> | 0.1 | 73 | 64 | 38 | 74 |
> | 0.3 | 70 | 62 | 38 | 76 |
> | 0.5 | 61 | 54 | 36 | 72 |
>
> The results indicate that increasing $\alpha$ leads to a decline in scores for Depth = 0% and 25%, which confirms the potential drawback of excessive regularization. This conclusion is also consistent with the claim in Section 5.4.
>
> ---
>
> Q5. One of the models looks very underfit in Figure 1a, perhaps worth investigating the training configuration.
>
> A5. The model that appears underfit in Figure 1(a) uses Alibi positional encoding. We indeed observe that the Alibi-based model converges more slowly than those using other positional encodings, likely due to the complex inductive bias introduced by Alibi.
>
> ---
>
> Q6. It could be interesting to see if different positional encoding schemes lead to different degrees of "long-short misalignment" in the predictable way.
>
> A6. Thank you for your thoughtful insight. Regarding the connection between positional encoding schemes and long-short misalignment, we observe that most modern LLMs rely on RoPE, often with modifications to its hyperparameters. However, as shown in Table 1, even among models using RoPE, length generalization ability varies significantly. For example, while GPT-J-6B, GPT-NeoX-20B, Llama2-7B, and Qwen-7B-8K all use RoPE, only Qwen-7B-8K demonstrates strong length generalization and low long-short misalignment. Furthermore, Yarn-Llama2-7B-8K and CLEX-Llama-4K, which employ modified versions of RoPE, also exhibit improved generalization and reduced misalignment. These observations suggest that while positional encoding choices influence long-short misalignment, they do not fully determine it.

---

### Decision · Program_Chairs · 2025-05-01

**Decision:**

Accept (poster)

**Comment:**

This paper studies how the output distributions of LLMs relates to their length generalization and long-context performance. The proposed "long-short misalignment" metric and a related regularization loss provides empirical evidence in both synthetic and real tasks. While some reviewers raised concerns around the motivation and theoretical framing, the authors addressed these well during rebuttal, including new results and clarifications. Overall, this paper offers some new insights into length generalization of LLMs, although the (1) comparisons/connections to other length generalization methods and (2) more clarity of the limitations of the proposed regularization loss could make this paper stronger.